# The human gut Firmicute *Roseburia intestinalis* is a primary degrader of dietary β-mannans

Sabina Leanti La Rosa [1], Maria Louise Leth[2], Leszek Michalak[1], Morten Ejby Hansen [2], Nicholas A. Pudlo[3], Robert Glowacki[3], Gabriel Pereira[3], Christopher T. Workman[2], Magnus Ø. Arntzen[1], Phillip B. Pope [1], Eric C. Martens[3], Maher Abou Hachem [2] & Bjørge Westereng[1]

β-Mannans are plant cell wall polysaccharides that are commonly found in human diets. However, a mechanistic understanding into the key populations that degrade this glycan is absent, especially for the dominant Firmicutes phylum. Here, we show that the prominent butyrate-producing Firmicute *Roseburia intestinalis* expresses two loci conferring metabolism of β-mannans. We combine multi-"omic" analyses and detailed biochemical studies to comprehensively characterize loci-encoded proteins that are involved in β-mannan capturing, importation, de-branching and degradation into monosaccharides. In mixed cultures, *R. intestinalis* shares the available β-mannan with *Bacteroides ovatus*, demonstrating that the apparatus allows coexistence in a competitive environment. In murine experiments, β-mannan selectively promotes beneficial gut bacteria, exemplified by increased *R. intestinalis*, and reduction of mucus-degraders. Our findings highlight that *R. intestinalis* is a primary degrader of this dietary fiber and that this metabolic capacity could be exploited to selectively promote key members of the healthy microbiota using β-mannan-based therapeutic interventions.

[1] Faculty of Chemistry, Biotechnology and Food Science, Norwegian University of Life Sciences, Aas N-1433 Norge, Norway. [2] Dept. of Biotechnology and Biomedicine, Danish Technical University, Kgs. Lyngby DK-2800, Denmark. [3] Department of Microbiology and Immunology, University of Michigan Medical School, Ann Arbor 48109 MI, USA. Correspondence and requests for materials should be addressed to B.W. (email: bjorge.westereng@nmbu.no)

The human gastrointestinal tract harbors an extremely dense and diverse microbial community, known as the gut microbiota[1]. In a mutually beneficial relationship, the gut microbiota supplies enzymes able to depolymerize dietary carbohydrates that cannot be hydrolyzed by human enzymes[2,3]. The monosaccharides generated are further fermented into host-absorbable metabolites, including the short-chain fatty acids butyrate, acetate, and propionate. In particular, butyrate produced by commensal bacteria serves as the main energy source for colonocytes[4,5] and it exhibits anti-carcinogenic, anti-inflammatory, and barrier protective properties in the distal gut[6–8]. The relevance of this metabolic output to human health has prompted increasing interest in intentionally modulating the composition of the gut microbiota to promote wellbeing and combat disease, e.g., by the use of prebiotics[9]. Established prebiotics have been traditionally developed based on their selective fermentation by bifidobacteria and lactobacilli generally regarded as conferring health benefits to the host. Notably, other potentially beneficial targets are the butyrate-producing Firmicutes[9].

*Roseburia* spp., together with *Faecalibacterium prausnitzii* and *Eubacterium rectale*, constitute a group of dominant butyrate-producing Firmicutes, estimated to account for 7–24% of the total bacteria in the healthy human colon[10,11]. Interest in *Roseburia* spp. has increased with reports that the abundance of these bacteria is reduced in individuals affected by inflammatory diseases[12–14] and colorectal cancer[15,16]. Complementary studies have shown that *Roseburia* spp. play an important role in the control of gut inflammatory processes[17], amelioration of atherosclerosis[18] and in the maturation of the immune system, primarily through the production of butyrate[19]. *R. intestinalis* preferentially colonizes the mucin layer[20,21] and this intimate association to the host may contribute to the local level of butyrate available for the colonic epithelial cells[22]. This species appears to be a specialist able to grow only on a few glycans[23,24] and has been recently shown to be a prominent xylan degrader in vitro[25] and in the healthy human colon[26].

β-Mannans are widespread in the human diet: they are widely used in food as thickening, stabilizing, and gelling agents[27] (glucomannan and galactomannan, Fig. 1). They are found in the endospermic tissue of nuts (homopolymeric mannan), coffee beans, coconut palm, tomato, and legume seeds (galactomannan)

(Fig. 1)[27–29], and play vital roles in the cell wall structure and as storage polysaccharides in plants. Notably, the structure of galactoglucomannan[29] from non-edible sources (softwood) shares striking similarities with that from edible sources (Fig. 1).

Prevalent Gram-negative *Bacteroides* spp. encode β-mannan polysaccharide utilization loci (PULs) and have been recently shown to utilize mannans[30–32]. Despite members of the Firmicutes phylum being numerically dominant in the gut, insight is lacking into the metabolic strategies adopted by these Gram-positive bacteria to utilize β-mannans.

Here, using a combination of microbiology,"omic" and enzymology approaches, we unravel the molecular mechanism evolved by *R. intestinalis* L1–82 to depolymerize β-mannans that are potentially available in the large intestine. Our findings show that *R. intestinalis* growth on β-mannan is dependent on the expression of a highly specific multi-modular cell attached endomannanase, an ATP-binding transporter and an intracellular enzyme cocktail through which linear and substituted manno-oligosaccharides are completely hydrolyzed to component monosaccharides for further metabolism. Using germ-free mice colonized with a model gut microbiota, we demonstrate that β-mannan alters the community composition, facilitating bacteria that have mannan degrading machineries. Besides extending the knowledge on the enzymatic basis of β-mannan-metabolism by members of the most numerous Firmicutes phylum, our results have implications for the design of targeted intervention strategies to manipulate the gut microbiota via supplementation of prebiotics to the diet to restore or improve health.

## Results

**Two multi-gene loci mediate β-mannan utilization.** *R. intestinalis* L1–82 grows efficiently on a variety of complex β-mannans as a sole carbon source (Fig. 2a), causing a concomitant acidification of the medium (Fig. 2b). To evaluate which fractions of β-mannan breakdown products are internalized, we analyzed the culture supernatants during *R. intestinalis* growth on AcGGM using high-performance anion-exchange chromatography with pulsed amperometric detection (HPAEC-PAD) (Supplementary Fig. 1a, b). Neither oligosaccharides nor monosaccharides accumulated in the stationary phase culture (Supplementary Fig. 1a, b), indicating that the bacterium possesses a highly efficient apparatus to cleave and import all the sugars derived from the breakdown of this complex glycan.

To examine the molecular basis underlying β-mannan utilization by *R. intestinalis*, we performed an RNA sequencing (RNAseq) transcriptional analysis during growth on konjac glucomannan (KGM), spruce acetylated galactoglucomannan (AcGGM) and glucose (Glc). The top 20 upregulated genes in β-mannan transcriptome encode a β-mannanase belonging to the glycoside hydrolase (GH) 26 family (GH26 according to the CAZy classification[33]), a solute binding protein (MnBP) and two permeases (MPP) of an ABC transporter, two phosphorylases (GH130), one epimerase (Mep), two β-glucosidases (GH3) and two carbohydrate esterases (CEs) (Fig. 2c and Supplementary Data 1). These genes are located in two loci, which were designated mannan-utilization locus large (MULL: ROSINTL182_05469–83) and mannan-utilization locus small (MULS: ROSINTL182_07683–85) (Fig. 2d). Among the MULL genes expression of a LacI-type transcriptional regulator, predicted glycosyl hydrolases belonging to GH113, GH36, GH1, and a phosphomutase also increased. The response was specific to β-mannan as no differential expression of these genes was observed during growth of *R. intestinalis* on galactose, a building block in mannan (Supplementary Table 1).

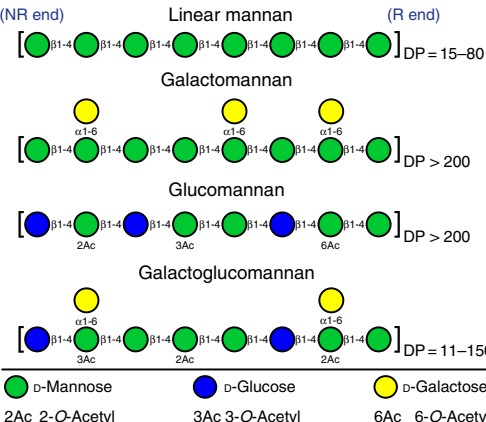

**Fig. 1** General structure of the main classes of β-mannan. Linear homopolymeric (upper structure) and linear heteropolymeric (lower three structures) β-mannans are shown. Monosaccharides (D-mannose, green circle; D-glucose, blue circle; D-galactose, yellow circle) and acetylations (2Ac, 2-O-Acetyl; 3Ac, 3-O-Acetyl; 6Ac, 6-O-Acetyl) are represented using the standard Consortium of Functional Glycomics symbols[67]. NR end, non-reducing end; R end, reducing end; DP, degree of polymerization

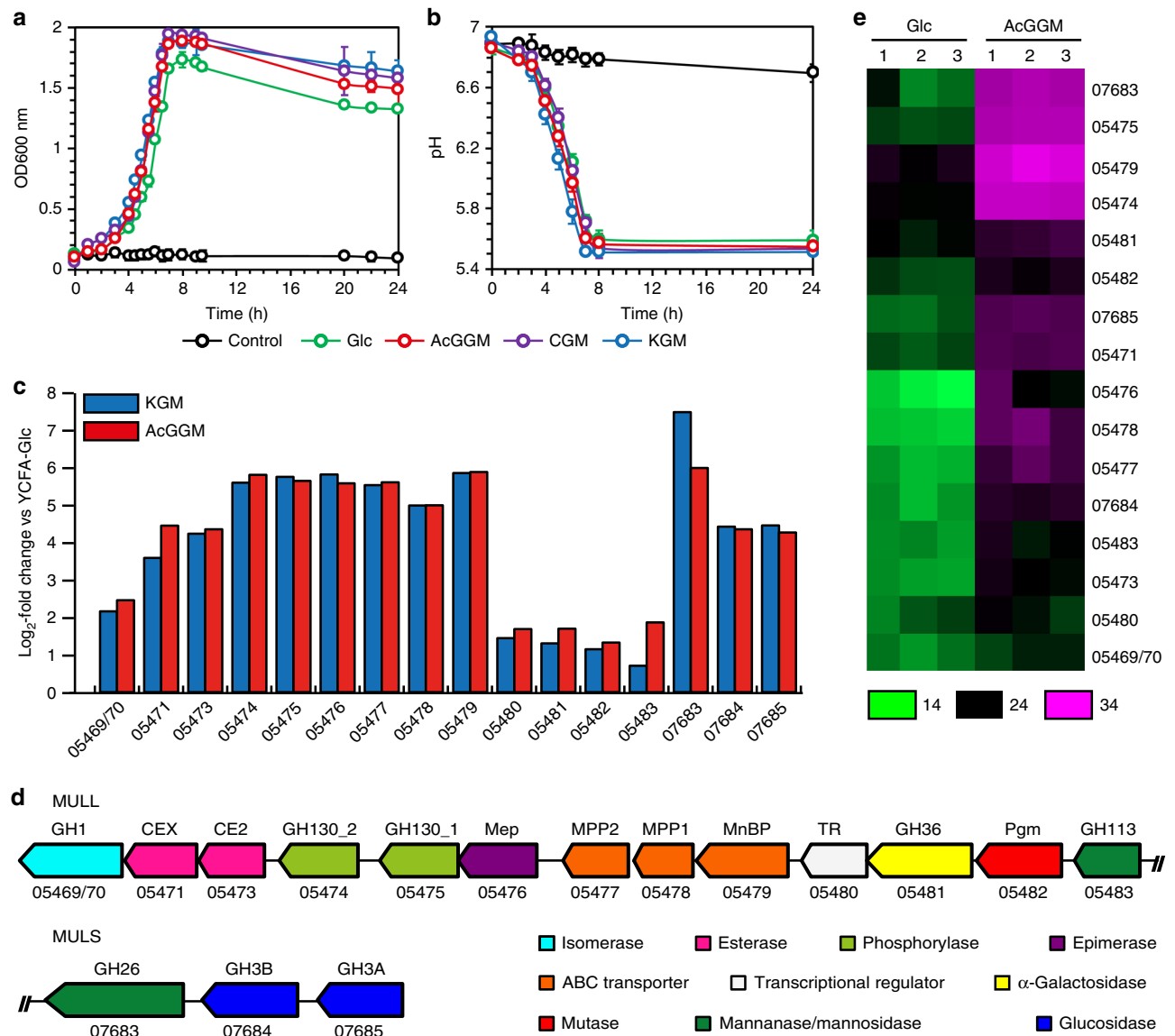

**Fig. 2** *R. intestinalis* upregulates several glycoside hydrolases, two carbohydrate esterases and an ABC-transporter during β-mannan consumption. **a** Growth curves of *R. intestinalis* in YCFA without carbon source (black) or supplemented with 0.5% of either glucose (Glc, green circles), KGM (blue circles), CGM (purple circles) or AcGGM (red circles). **b** pH measurements during *R. intestinalis* growth on Glc and β-mannans. In **a** and **b**, each point on the curves represent the average of three independent experiments. Error bars represent standard deviations (s.d.). **c** RNA expression profile of putative β-mannan utilization genes during *R. intestinalis* L1–82 growth in YCFA supplemented with 0.5% KGM (blue bars) or AcGGM (red bars). The Log₂-fold change relative to cells cultured on YCFA-Glc is shown on the *y*-axis while the *x*-axis shows the putative genes involved in β-mannan catabolism. **d** Genomic organization of the large and small β-mannan utilization loci (MULL and MULS, respectively) from *R. intestinalis*. Genes with similar predicted functions are coded by the same color. **e** Heat map showing the proteomic detection of relevant proteins with predicted β-mannan utilization functions in triplicate samples (1−3) grown on YCFA-Glc and YCFA-AcGGM. Colors represent protein intensity expressed as Log₂ of LFQ values; the color gradient ranges from 14 (green) to 34 (magenta), with black indicating 24. In **c**–**e**, locus tag numbers ROSINTL182_XXXXX are abbreviated with the last numbers after the hyphen

Proteomic analysis under the same growth conditions corroborated the RNAseq results; indeed, proteins encoded by the genes located in MULL and MULS were abundant in the AcGGM samples compared to the glucose samples (Fig. 2e, Supplementary Data 2).

We carried out a comparative genomic analysis to establish the distribution of β-mannans utilization loci equivalent to the identified MULL and MULS in other representative Roseburia *spp*. and Clostridium cluster XIVa members. The results showed that *R. faecis* and *R. hominis* shared an overall MULL and MULS organization with that of *R. intestinalis* (Supplementary Fig. 2, Supplementary Table 2), suggesting that the utilization of β-mannan is shared by these three *Roseburia* spp. However, the lack

of the critical GH26 endomannanase, required to break down mannan (see later results for *R. intestinalis* β-mannanase *Ri*GH26), is likely to render *R. hominis* only able to metabolize manno-oligosaccharides. Orthologous mannan utilization loci were identified in specific members of the Clostridium cluster XIVa, although a similar organization and complete conservation of all MULL and MULS components was not observed (Supplementary Fig. 2).

**Degradation of the β-mannan backbone.** *Ri*GH26, (locus tag: ROSINTL182_07683), is a predicted extracellular modular β-mannanase comprising a carbohydrate binding module of family

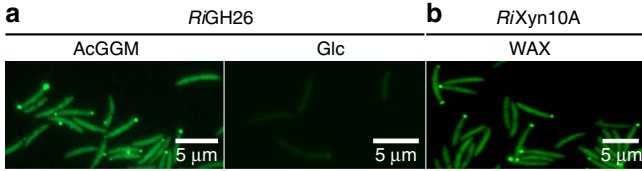

**Fig. 3** Cellular location of the endomannanase *Ri*GH26. **a** Fluorescent microscopy images of *R. intestinalis* cells cultured on AcGGM or Glc and incubated with polyclonal antibodies raised against the recombinant endomannanase *Ri*GH26. Glucose-grown cells exhibit a low intensity fluorescence signal; this is consistent with the results of the proteomics data showing that, when the organism is cultured on glucose, *Ri*GH26 is expressed at basal levels. **b** Fluorescent microscopy images of *R. intestinalis* cells grown on WAX (positive control) and incubated with antibodies raised against the known surface endoxylanase *Ri*Xyn10A[25]. Localization microscopy images are representative data from two biological duplicates

**Table 1 Binding parameters of *Ri*CBM27 and *Ri*CBM23 to manno-oligosaccharides and cello-oligosaccharides**

|  | $K_d$ (µM) | |
| --- | --- | --- |
| Ligand | *Ri*CBM27 | *Ri*CBM23 |
| $M_3$ | 1593 ± 30 | 230 ± 20 |
| $M_4$ | 658 ± 20 | 130 ± 50 |
| $M_5$ | 321 ± 20 | 198 ± 70 |
| $M_6$ | 165 ± 10 | 205 ± 40 |
| $Glc_4$ | No binding | No binding |
| $Glc_6$ | No binding | No binding |

Binding was determined by SPR. Values show the means and standard deviations of at least two independent experiments
$K_d$ dissociation constant

**Table 2 Thermodynamic binding parameters of *Ri*MnBP to linear and decorated manno-oligosaccharides**

| Ligand | $K_d$ (µM) | $\Delta G$ (kcal mol$^{-1}$) | $\Delta H$ (kcal mol$^{-1}$) | $-T\Delta S$ (kcal mol$^{-1}$) | $n$ |
| --- | --- | --- | --- | --- | --- |
| $M_3$ | 2.62 | −7.6 | −33.2 | 25.6 | 0.7 |
| $M_4$ | 3.89 | −7.4 | −28.6 | 21.2 | 0.7 |
| $M_5$ | 2.55 | −7.7 | −21.8 | 14.1 | 0.8 |
| $M_6$ | 33.75 | −6.2 | −17.8 | 11.6 | 0.5 |
| $M_4Ac_2$ | 25.65 | −6.3 | −21.9 | 15.6 | 0.9 |
| $M_5Ac_2$ | 23.53 | −6.3 | −20.2 | 13.9 | 0.8 |

Binding was measured by ITC. Data are means of two independent titrations
$K_d$ dissociation constant, $\Delta G$ Gibbs free energy, $\Delta H$ enthalpy, $-T\Delta S$ entropy, $n$ binding stoichiometry

27 (CBM27), a catalytic module of GH26 followed by a CBM23 (Supplementary Fig. 3a). Furthermore, two C-terminal Ig-like domains and a proline-glycine rich region likely mediate cell attachment[34] and binding within the cell wall[35]. The extracellular localization of *Ri*GH26 was corroborated experimentally by immunofluorescence microscopy (Fig. 3). *Ri*GH26 exhibited activity toward decorated mannans including KGM, carob galactomannan (CGM) and AcGGM (Fig. 4a and Supplementary Fig. 3b), generating linear and substituted manno-oligosaccharides. The enzyme was active on mannopentaose ($M_5$) and mannotetraose ($M_4$) but not mannobiose ($M_2$) (Supplementary Fig. 3c). Overall, the product profiles suggest capability of endo-action and indicates that *Ri*GH26 targets large polymers and can accommodate the galactose and acetyl decorations present in these substrates. Further analysis indicated that *Ri*GH26 is a potent enzyme as, when used at the concentration of 10 nM, it was able to hydrolyze high concentrations of spruce AcGGM (50 mg ml$^{-1}$) into oligosaccharides in 1 h at standard assay conditions (Supplementary Fig. 3d). No detectable activity was measured when *Ri*GH26 was incubated with linear cello-oligosaccharides, birch xylan, curdlan, lichenan or barley derived β-glucan, thus confirming the specificity of *Ri*GH26 towards β-mannan (Supplementary Fig. 3c).

BlastP searches showed that homologs of *Ri*GH26, including the two predicted carbohydrate binding modules CBM27 and CBM23, were exclusively found in β-mannanase encoded by Firmicutes belonging to various other members of the Clostridium cluster XIVa (Supplementary Fig. 4, Supplementary Table 3–5). To investigate the biochemical properties of the two modules, *Ri*CBM27 and *Ri*CBM23 were expressed in *Escherichia coli* and their capacities to bind to a range of different soluble cello-oligosaccharides and manno-oligosaccharides were evaluated using surface plasmon resonance (SPR). Recombinant *Ri*CBM27 and *Ri*CBM23 bound only manno-oligosaccharides (Table 1), but differed in their binding profiles. Similar to a previously described GH26-associated CBM27[36], *Ri*CBM27 preferred mannohexaose ($M_6$) ($K_d$ = 165 ± 10 µM, two independent experiments, ± indicates standard deviation), (Table 1, Supplementary Fig. 5a) and its affinity dropped for ligands smaller than a tretrasaccharide (Table 1). By contrast, *Ri*CBM23 was selective for shorter oligosaccharides with its highest affinity for $M_4$ ($K_d$ = 130 ± 50 µM, two independent experiments) (Table 1, Supplementary Fig. 5b), although mannotriose ($M_3$) was also bound with good affinity (Table 1).

**Internalization of break-down products from β-mannan.** Within the MULL cluster, the three genes (ROSINTL182_05477 – ROSINTL182_05479) that encode an ATP-binding cassette

(ABC) importer were shown to exhibit the highest level of increased expression during growth on β-mannan (and when compared to growth on glucose). The thermodynamic binding parameters of the ABC-transporter associated solute binding protein, *Ri*MnBP, to linear and substituted manno-oligosaccharides were determined using isothermal titration calorimetry (ITC). *Ri*MnBP bound a range of unsubstituted manno-oligosaccharide with a preference for $M_5$ ($K_d$ of 2.55 µM) followed by $M_3$ and $M_4$ (Table 2, Supplementary Fig. 6). Acetylations had a marginal effect on the binding affinities, thus providing evidence that these fragments are efficiently captured by the transport protein. Overall, these results support the predicted role of *Ri*MnBP in the uptake of manno-oligosaccharides generated by *Ri*GH26, showing optimal affinity for undecorated or acetyl substituted ligands with a degree of polymerization (DP) of 4−5.

**Decomposition of internalized β-manno-oligosaccharides.** The affinity of the solute binding protein *Ri*MnBP to manno-oligosaccharides and the predicted intracellular location of the debranching and exo-acting enzymes is consistent with a hierarchical degradation of the internalized manno-oligosaccharides following extracellular degradation of the β-mannan polymers by *Ri*GH26.

The ROSINTL182_05471 (*Ri*CEX) and ROSINTL182_05473 (*Ri*CE2) gene products possess SGNH hydrolase-type esterase domain signatures[37]. Comparison to previously characterized CEs revealed that *Ri*CE2 showed 25–30% identity to a CE2 from *Clostridium thermocellum*[38] and the acetyl xylan esterase Axe2C of *Cellvibrio japonicus*[38]. In contrast, *Ri*CEX did not display

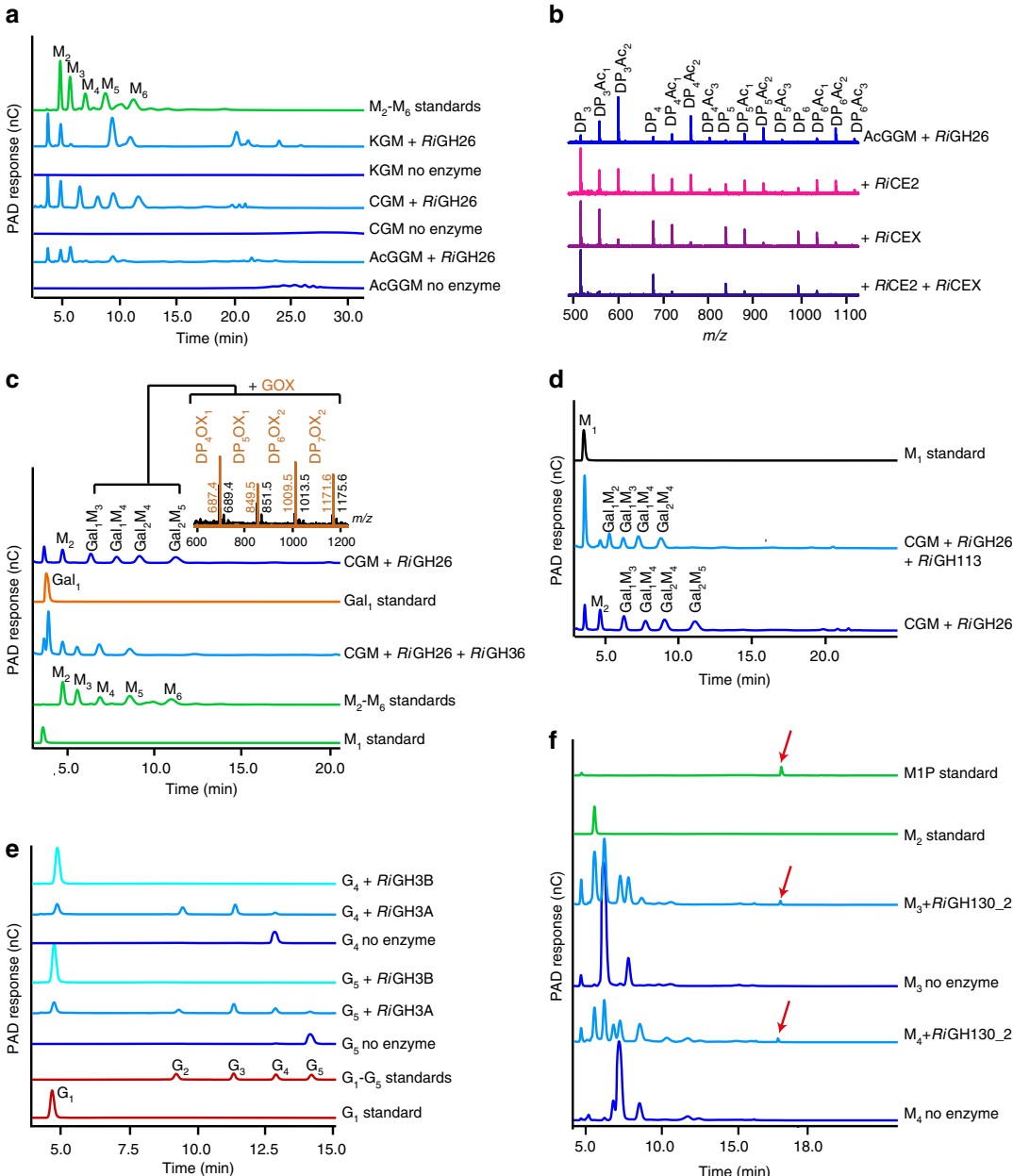

**Fig. 4** Cleavage of the β-mannans backbone, removal of the side chains and further depolymerization of the resulting linear manno-oligosaccharides. **a** HPAEC chromatograms showing the oligosaccharide products after overnight digestion of KGM, CGM and AcGGM with *Ri*GH26. Samples were analyzed with the following manno-oligosaccharides as external standards: M₂, mannobiose; M₃, mannotriose; M₄, mannotetraose; M₅, mannopentaose; M₆, mannohexaose. **b** MALDI-TOF analysis of *Ri*GH26-digested AcGGM incubated with either *Ri*CE2, *Ri*CEX or both enzymes. Peaks are labeled by DP and number of acetyl groups (Ac). **c** HPAEC chromatograms showing products generated from CGM pre-digested with *Ri*GH26 and subsequently treated with *Ri*GH36. Assignments for peaks not occurring in the standard samples are based on comparison with the product profiles obtained by MALDI-TOF MS of *Ri*GH26-digested CGM (black spectrum) treated with galactose oxidase (GOX; brown spectrum). GOX converts a galactose residue in the oligosaccharides into its corresponding aldehyde, giving a mass-to-charge ratio (*m/z*) of -2. All assigned masses are sodium adducts. Abbreviations: Ox, oxidation; Gal₁, galactose; Gal₁M₃, galactosylmannotriose; Gal₁M₄, galactosylmannotetraose; Gal₂M₄, digalactosylmannotetraose; Gal₂M₅, digalactosylmannopentaose. **d** Product profiles from *Ri*GH26-digested CGM degradation experiments with *Ri*GH113 analyzed by HPAEC-PAD. The release of mannose confirms the exo-activity of *Ri*GH113. **e** HPAEC-PAD traces showing activity of *Ri*GH3A or *Ri*GH3B towards G₅ and G₄ with the corresponding controls (no enzyme). Product profiles at various time points during the reaction are shown in Supplementary Fig. 6. Taken together, the data show that *Ri*GH3B is able to hydrolyze completely both tetramers and pentamers, producing glucose. *Ri*GH3A shows exo-activity towards both substrates that are converted slowly to glucose and a mixture of cello-oligosaccharides. Samples were analyzed with the following cello-oligosaccharides as external standards: glucose, G₁; cellobiose, G₂; cellotriose, G₃; cellotetraose, G₄; cellopentaose, G₅. **f** Chromatograms showing products generated upon incubation of *Ri*GH130_2 with M₄ and M₃. The M1P released (red arrow) was identified by co-migration with the appropriate standard. In all panels, the data displayed are representative of at least three biological triplicates

significant relatedness to other characterized CE catalogued in the CAZy database[33], which excluded $Ri$CEX from being classified in any of the 16 CE families. $Ri$CEX and $Ri$CE2 showed mannan acetyl esterase activity on a mixture of oligosaccharides generated via $Ri$GH26 hydrolysis of AcGGM (Fig. 4b). $Ri$CE2 partially removed acetyl groups from the acetylated oligosaccharide substrate (Fig. 4b). $Ri$CEX deacetylated the substrate mainly to free and monoacetylated oligosaccharides (Fig. 4b). These results indicate that $Ri$CEX has a preference for oligosaccharides with a degree of acetyl substitution ≥2, but is less efficient on mono-substituted substrates. At the same time, it suggests that an acetyl group present at a specific position ($O$-2 or $O$-3) is resistant to enzymatic deacetylation by $Ri$CEX. The combination of $Ri$CEX and $Ri$CE2 resulted in the almost complete deacetylation of the manno-oligosaccharides, indicating a cooperative interaction of the two esterases (Fig. 4b).

$Ri$GH36 released galactose from internally substituted CGM and AcGGM after the treatment with the $Ri$GH26 β-mannanase (Fig. 4c and Supplementary Fig. 7). Interestingly, $Ri$GH36 released galactose from CGM-endomannanase products with 100% efficiency (Fig. 4c, Supplementary Fig. 8a) as no oxidized product could be observed after treatment of these samples with galactose oxidase. The enzyme exhibited limited activity on large polymers (Supplementary Fig. 8b) consistent with the activity on internalized oligosaccharides in vivo. Similarly, α-galactosidase activity increased after de-acetylation of the oligosaccharides (Supplementary Fig. 8c, d). Beside cleaving single internal galactose residues from manno-oligosaccharides, this enzyme was capable of removing α-1,6-galactose from the reducing-end of a substituted manno-oligosaccharide (Supplementary Fig. 8e) and from an oligosaccharide containing two consecutive substitutions (Supplementary Fig. 8f). Corroborating these results, $Ri$GH36 cleaved galactose decorations from endomannanase products of highly substituted guar gum galactomannan (Supplementary Fig. 8b).

Sequence searches showed that the protein encoded by ROSINTL182_05483 (MULL, $Ri$GH113) exhibited 40% identity to the only characterized enzyme from this family, the endo-β-mannanase AaManA from *Alicyclobacillus acidocaldarius*[39] (Supplementary Fig. 9a). Alignment of $Ri$GH113 and AaManA showed that the catalytic and substrate interacting residues are conserved (Supplementary Fig. 9a). When $Ri$GH113 was assayed for activity on linear manno-oligosaccharides, it revealed an ability to cleave linear manno-oligosaccharides to yield mannose and $M_2$ (Supplementary Fig. 9b). Strikingly, time-course analysis of $Ri$GH113 activity revealed that this enzyme displays a different sub-specificity by hydrolyzing manno-oligosaccharides to mannose and a minor amount of $M_2$ (Supplementary Fig. 9c). After overnight incubation with $Ri$GH113, $M_2$ was partially degraded to mannose (Supplementary Fig. 9d), confirming the exo-mannosidase activity as opposed to the endo-fashion cleavage reported for the AaManA. The substituted manno-oligosaccharides galactosylmannobiose ($Gal_1Man_2$) and digalac-tosylmannopentaose ($Gal_2Man_5$) were hydrolyzed to a lesser extent than non-substituted substrates (Supplementary Fig. 9e); no activity could be detected on $Gal_1Man_2$ while $Gal_2Man_5$ was hydrolyzed to yield mannose and digalactosylmannotetraose ($Gal_2Man_4$), which was resistant to further hydrolysis. When the reducing end of manno-oligosaccharides was blocked (Supplementary Fig. 10a–d), no $Ri$GH113 activity could be detected demonstrating that this enzyme possesses a previously unknown reducing end mannose-releasing exo-oligomannosidase activity. Consistent with the view that $Ri$GH113 is an intracellular enzyme, release of mannose was detected after incubation of the enzyme with $Ri$GH26-generated CGM-oligosaccharides (Fig. 4d). The closest homologs of this enzyme are encoded by Clostridium

cluster XIVa strains and a range of Firmicutes (Supplementary Fig. 10e).

Product analysis of end point assays and a time course study revealed that both $Ri$GH3A (ROSINTL182_07684) and $Ri$GH3B (ROSINTL182_07685) were β-glucosidases, with redundancy in structure (Supplementary Fig. 11a, b), active on linear cello-oligosaccharides (Fig. 4e). $Ri$GH3B completely hydrolyzed cellotetraose ($G_4$) and cellopentaose ($G_5$) into glucose monomers, whereas $Ri$GH3A released glucose and a range of oligosaccharides with lower efficiency compared to that of $Ri$GH3B (Supplementary Fig. 11c, d). Neither of these enzymes were active on manno-oligosaccharides (Supplementary Fig. 11e, f). While $Ri$GH3B was able to hydrolyze glucosylmannose ($G_1M_1$) and, partially, mannosylglucose ($M_1G_1$) into monomers (Supplementary Fig. 11f), $Ri$GH3A displayed activity only towards $G_1M_1$. No activity was detected on polymeric KGM (Supplementary Fig. 12a), while glucose was released after incubation of both $Ri$GH3A and $Ri$GH3B with $Ri$GH26-generated KGM–hydrolysate (Supplementary Fig. 12b). Importantly, the latter results demonstrate that $Ri$GH26 can accept a glucose moiety at the subsite +1, generating oligosaccharides with a glucose residue at the non-reducing end.

Recombinant $Ri$GH130_2 (MULL, ROSINTL182_05474) phosphorolyzed $M_4$ into $M_3$, $M_2$ and mannose-1-phosphate (M1P) while $M_3$ was processed to $M_2$ and M1P (Fig. 4f). The enzyme was inactive on cello-oligosaccharides (Supplementary Fig. 13). $Ri$GH130_2 was functional only in the presence of inorganic phosphate, confirming that $Ri$GH130_2 is a mannosyl-phosphorylase.

**Catabolism of mannobiose and mannosylglucose units**. The concerted action of the MULL and MULS encoded enzymes described above on the oligosaccharides generated by $Ri$GH26, suggest an intracellular accumulation of $M_2$. Hydrolysis of this product into monosaccharides is accomplished through the action of two enzymes encoded by the co-transcribed genes ROSINTL182_05476 ($Ri$Mep) and ROSINTL182_05475 ($Ri$GH130_1).

$Ri$Mep was active on $M_2$ and cellobiose ($G_2$), releasing $M_1G_1$ and $G_1M_1$, respectively (Fig. 5a). These data show that $Ri$Mep is an enzyme active on the reducing end sugar and catalyzes the conversion of disaccharide substrates to the corresponding C2 epimer. This enzyme exhibited epimerization activity not only for the substrate but also for the product as, under high enzyme amount and long reaction time, it was able to convert $M_1G_1$ and $G_1M_1$ to $M_2$ and $G_2$, respectively (Supplementary Fig. 14a). In addition, $Ri$Mep exhibited epimerization activity towards oligosaccharides with a DP > 2 but not on mono-saccharides (Supplementary Fig. 14b).

ROSINTL182_05475 encodes a specific mannosylglucose phosphorylase belonging to the GH130 subfamily 1[40]. $Ri$GH130_1 was inactive on $G_1M_1$ and oligosaccharides with a DP ≥ 2 (Supplementary Fig. 15). $Ri$GH130_1 displayed activity only towards $M_1G_1$ in the presence of inorganic phosphate, releasing glucose and M1P (Fig. 5b, c).

**Catabolism of phosphorolysis products**. $Ri$Pgm catalyzes the interconversion of M1P and mannose 6-phosphate (M6P) (Fig. 5d). In addition, the enzyme displayed activity also against D-glucose 1-phosphate (G1P) yielding D-glucose 6-phosphate (G6P) (Supplementary Fig. 16a), thus indicating that $Ri$Pgm is a phosphomannomutase (PMM)/phosphoglucomutase (PGM) which can use either glucose or mannose as substrate. Consistent with the presence of a predicted magnesium-binding loop

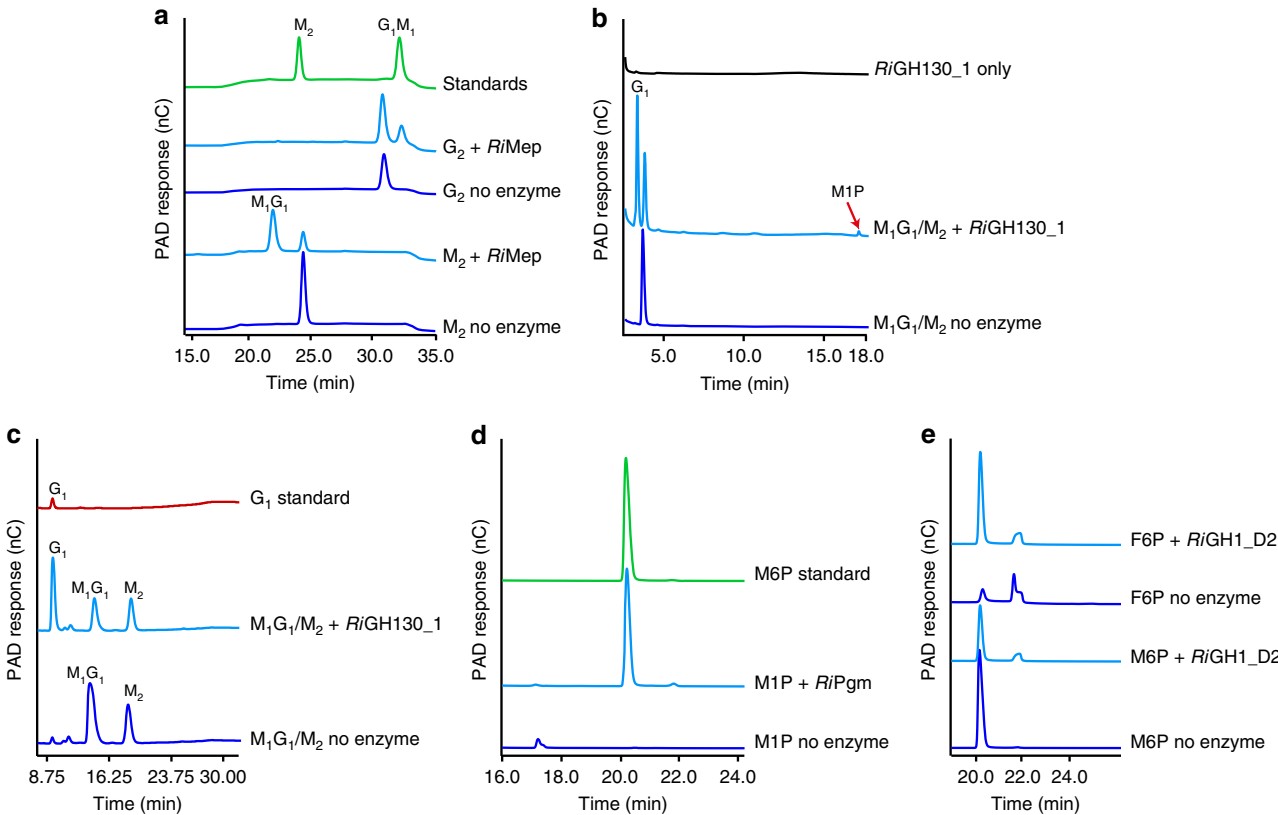

**Fig. 5** Enzymes for catabolism of mannobiose, mannosylglucose and monosaccharides deriving from complex β-mannan degradation. **a** HPAEC-PAD traces showing the epimerization of $M_2$ and $G_2$ by RiMep to release $M_1G_1$ and $G_1M_1$, respectively. **b, c** HPAEC-PAD of phosphorolysis reactions of RiGH130_1 using premixed $M_1G_1$ plus $M_2$ from Megazyme. The reaction products were then analyzed using an HPAEC method designed for the separation of **b** phosphorylated manno-oligosaccharides or **c** mono-saccharides and di-saccharides. M1P, indicated with a red arrow, and $G_1$ peaks in **b** are marked according to the standards. **d** HPAEC-PAD analysis of RiPgm-catalyzed conversion of M1P to M6P. The M6P released was identified by co-migration with the M6P standard. **e** Activity of RiGH1_D2 on M6P and F6P analyzed by HPAEC-PAD

(Supplementary Fig. 16b), the RiPgm-mediated catalytic activity was detected only when MgCl₂ was present in the reaction.

ROSINTL182_05469/70 encodes a predicted bi-functional protein consisting of an N-terminal glucosidase domain (RiGH1_D1, aa 1–246) and a C-terminal family GH1 isomerase domain (RiGH1_D2, aa 247–768). RiGH1_D1 shares 44% identity to the previously characterized β-glucosidase TmGH1 from *Thermotoga maritima*[41]. The recombinant RiGH1_D1 displayed no catalytic activity against all the tested substrates, including $G_4$, $M_4$, $M_5$, M6P, G6P, and fructose 6-phosphate (F6P). Thus, RiGH1_D1 functional significance is currently unclear. RiGH1_D2 is a phosphomannose isomerase catalyzing the interconversion of M6P into F6P (Fig. 5e).

**R. intestinalis competes with Bacteroides for β-mannans.** The ability of *R. intestinalis* to capture, breakdown β-mannan and efficiently internalize manno-oligosaccharides may increase its fitness to compete with other resident β-mannan degraders, including the glycan generalist *Bacteroides*[30]. To test this hypothesis, we performed in vitro co-cultivation of *R. intestinalis* and the efficient β-mannan degrader *Bacteroides ovatus* ATCC 8483[30]. Both bacteria showed similar growth rates in individual cultures supplemented with AcGGM (Fig. 6a). Population estimates using qPCR indicated that, in the mixed cultures, both *B. ovatus* and *R. intestinalis* grew well during the exponential growth phase, suggesting that the bacteria shared the available carbon source and maintained coexistence. (Fig. 6b). During the

stationary phase, when glycan availability is limited, the mean relative abundance of *R. intestinalis* and *B. ovatus* in the culture was approximately 72.5% versus 27.5%, respectively. In contrast, *R. intestinalis* showed slow growth on mannose (Fig. 6c) and was outcompeted when co-cultured in this carbon source with *B. ovatus* (Fig. 6d).

**R. intestinalis responds rapidly to β-mannan supplementation.** To elucidate whether dietary supplementation of β-mannan can result in expansion of key gut bacteria able to utilize this hemi-cellulose, germfree mice were colonized with a synthetic micro-biota composed of 14 sequenced strains of human commensal gut bacteria[42]. Colonized mice were fed a high-fiber diet for 14 days before being switched to a series of diet regimes with a varying amount of AcGGM (Fig. 6e). Overall, the levels of four species (*R. intestinalis*, *Bacteroides uniformis*, *B. ovatus* and *Marvinbryantia formatexigens*) gradually increased at both AcGGM doses (Fig. 6f–i) and these strains were able to suppress the bacteria foraging on the glyco-protein rich mucus layer (*Akkermansia muciniphila*, *Bacteroides caccae*, *Bacteroides thetaiotamicron*, *Bacteroides intestinihominis*) (Fig. 6e and j–m) and the amino acids degraders (*E. coli*, *Clostridium symbiosum* and *Collinsella aerofaciens*) (Fig. 6e and n). Upon feeding of a fiber-deficient diet, the fecal bacterial abundance of the mucin-eroding bacteria, the sulfate-reducer *Desulfovibrio piger* and the three amino acid degraders (Fig. 6e) rapidly increased with a corresponding decline of the fiber-degrading species.

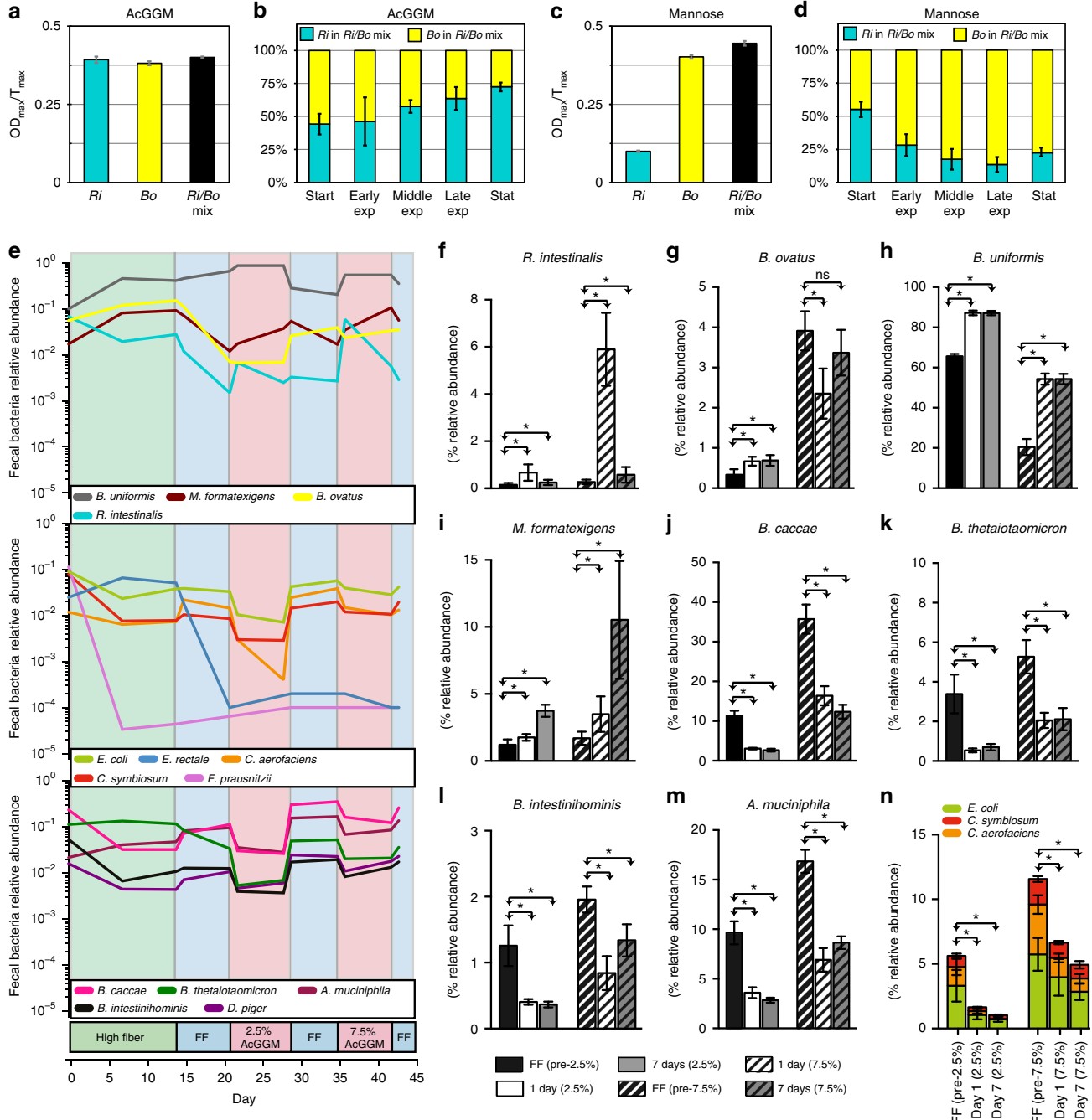

**Fig. 6** *R. intestinalis* and *B. ovatus* co-culture experiments and in vivo modulation of a synthetic human gut microbiota via AcGGM. **a**, **c** Growth rates of mono- and mixed cultures of *R. intestinalis* L1–82 (*Ri*) and *B. ovatus* ATCC 8483 (*Bo*) on either AcGGM or mannose. Growth rate is defined as the maximum increase in absorbance at 600 nm ($OD_{max}$) divided by the time ($T_{max}$, in hours) to reach the maximum growth. **b**, **d** In vitro competition experiment with *R. intestinalis* L1–82 and *B. ovatus* ATCC 8483 on either AcGGM or mannose as sole carbon source. The pH of the stationary phase cultures after growth on either AcGGM or mannose was 5.8 ± 0.16 and 5.6 ± 0.11 (two biological triplicates, ± indicates the s.d.), respectively, thus showing that the results are not due to differences in acid sensitivity between the two strains. The relative abundance of the bacteria for each different phases of growth was determined by quantitative PCR of species-specific vs universal primers targeting the 16 S rRNA genes. In **a**–**d**, the histogram bars show the mean of two biological replicates, with three independent measurements per replicate. Error bars represent s.d. Abbreviations: Early exp, early exponential phase; Middle Exp; middle exponential phase; Late exp, late exponential phase; Stat, stationary phase. **e** Relative abundance of bacteria in fecal samples from germfree mice colonized with a synthetic human microbiota. Mice were shifted from a fiber-free (FF) diet to varying amounts (2.5% and 7.5% w/w) of AcGGM over time. Data are average of seven mice. **f**–**i** Relative abundance of individual β-mannan-degrading bacteria and **j**–**m** mucus-degraders. **n** Additive relative abundances of three amino acids degraders. In **f**–**n** histogram bars show the average of seven biological replicates while error bars represent s.d. *P*-values were calculated by two-tailed Student's *t* test. An asterisk (*) indicate a statistically significant difference (*P* < 0.05) in the relative abundance of each bacterium compared to that of the specific pre-FF diet. ns, not significant (*P* ≥ 0.05)

## Discussion

β-Mannans are widely present in the human diet as constituents of hemicellulose in beans, some nuts and food additives, but are recalcitrant to intestinal digestion by host enzymes. These glycans instead serve as a carbon source for mannanolytic bacteria in the distal gastrointestinal tract, primarily Firmicutes and Bacteroidetes. Recent studies have characterized a few enzymes encoded by two polysaccharide utilization loci (PULs) implicated in the metabolism of galactomannan in *B. ovatus*[30,43] and homopolymeric mannan in *Bacteroides fragilis*[32]. To date, a full understanding of β-mannan utilization by Firmicutes, however, remains underexplored. The human gut butyrate-producing Firmicute *R. intestinalis* has previously been shown to utilize galactomannan and glucomannan as a carbon source[42] and possesses predicted genes for the utilization of these substrates[24]. However, no data are available relating the mannanolytic activity at a biochemical level. In this study, we show that two conserved loci, MULL and MULS, collectively provide *R. intestinalis* the capacity to depolymerize this plant polysaccharide. Detailed biochemical studies of the encoded enzymes allowed us to construct a model of sequential action for the mannan utilization system encoded by MULL-MULS (Fig. 7). The *Ri*GH26 and the mannan ABC uptake system components *Ri*MnBP/*Ri*MPP1/*Ri*MPP2 transcripts and proteins were the most upregulated in both the RNA sequencing and proteomic analyses, respectively (Fig. 2c, e). This highlights the crucial role of this endomannanase and the ABC transport system in the β-mannan metabolic pathway. *Ri*GH26 is the only enzyme displayed on the cell surface (Fig. 3), allowing direct access to the intact β-mannan polymers through dynamic capture mediated by two appended carbohydrate binding modules (*Ri*CBMs). The SPR data showed that *Ri*CBM23 displays ~7- and 5-fold higher affinity for $M_3$ and $M_4$, respectively, than *Ri*CBM27, suggesting that the two CBMs play different roles to mediate binding of *Ri*GH26 to mannans. The *Ri*CBMs' $K_d$ values for the preferred manno-oligosaccharides were in the 100−200 μM range (Table 1). This moderate affinity to the bound substrate constitutes an advantage as it has lower impact on the catalytic activity compared to canonical counterparts from other organisms, and suggests an evolutionary adaptation of *R. intestinalis* to compete with other microbial enzymes

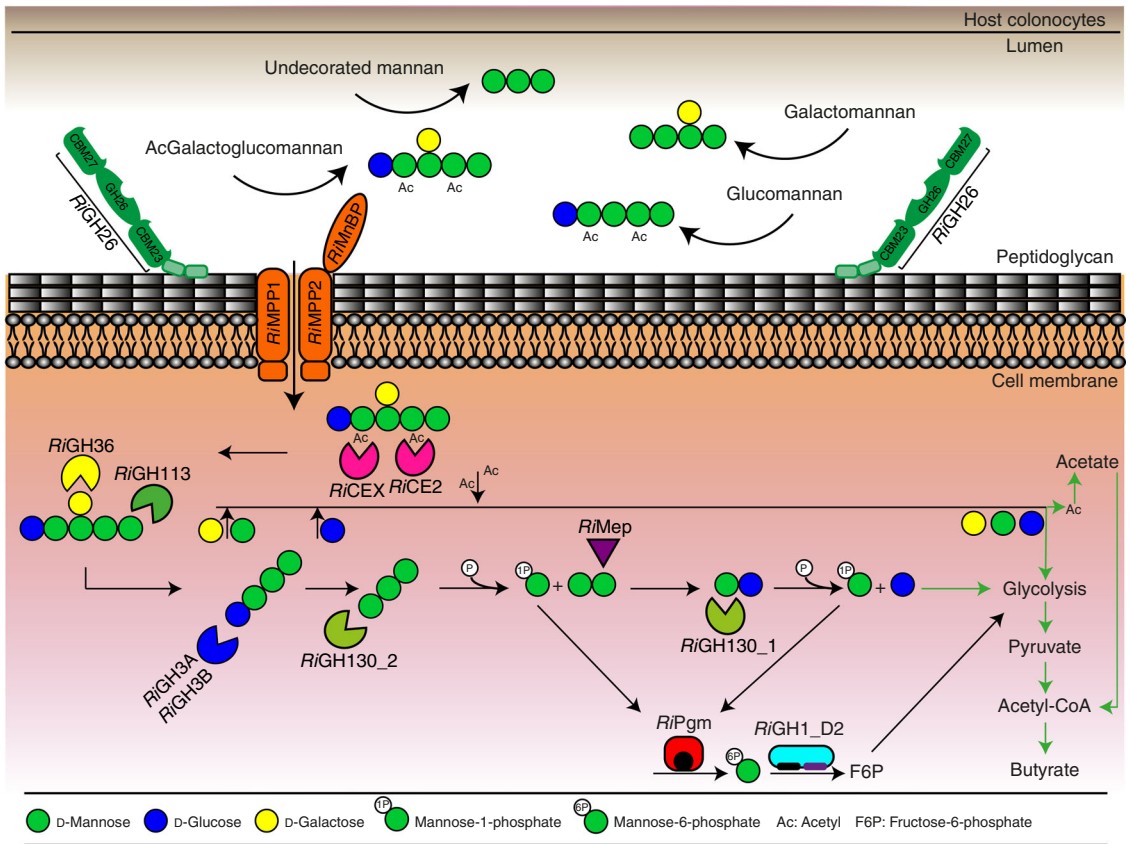

**Fig. 7** Model for the degradation and utilization of complex β-mannans in *R. intestinalis*. Intracellular degradation of an acetylated galactoglucomanno-oligosaccharide is used as an example. Sugars are represented as in Fig. 1. Initial depolymerization of acetylated galactoglucomannan (AcGalactoglucomannan) occurs at the outer surface of *R. intestinalis* by the activity of *Ri*GH26 (green). The extracellular recruitment of β-mannan is facilitated by interactions with CBM27 and CBM23. Import of products occurs through the ABC transporter *Ri*MnBP/RiMPP1/RiMPP2 (orange). Within the cytoplasm, the acetyl and galactosyl decorations are removed by the two acetyl esterases *Ri*CE2 and *Ri*CEX (pink) and the α-galactosidase *Ri*GH36 (yellow). The two β-glucosidases *Ri*GH3A and *Ri*GH3B (blue) release glucose from the non-reducing end of the β-manno-oligosaccharide. In addition, the reducing end mannose-releasing exo-oligomannosidase *Ri*GH113 (green) can catalyze the removal of mannose units from the decorated manno-oligosaccharides until it reaches a galactosyl substituent at the subsite −1. Once de-ornamented, the β-manno-oligosaccharides are saccharified by the exo-acting *Ri*GH130_2 (light green) with accumulation of $M_2$. The $M_2$ undergoes subsequent epimerization and phosphorolysis by the concerted activity of *Ri*Mep - *Ri*GH130_1 (light green), with release of glucose and M1P. These end products enter the glycolytic pathway either directly (for glucose) or after being converted into M6P and F6P by the phosphomannose mutase *Ri*Pgm (red) and the isomerase *Ri*GH1_D2 (turquoise, purple domain). Released mannose is converted to M6P by a hexokinase and processed as described above. Galactose enters glycolysis after conversion to G1P via the Leloir pathway. The pyruvate generated from glycolysis is converted to acetyl-CoA and then butyrate. Black arrows show reactions demonstrated in this study. Green arrows indicate previously demonstrated steps for the generation of butyrate from monosaccharides fermentation[68] by *R. intestinalis*

with canonical higher-affinity CBMs, but with reduced catalytic rates[44]. Reliance on multi-modular cell-wall anchored enzymes is a common feature in Firmicutes;[45] consistently, *Ri*GH26 organization was primarily found in β-mannanase from other *Roseburia* species and members of the Clostridium cluster XIVa (Supplementary Fig. 4, Supplementary Table 3–5). Multiplicity of CBMs provides a contrast with the system for mannan metabolism in *Bacteroides ovatus*[30,43], where the binding and catalytic activity are distributed between two surface located binding proteins and the single domain mannanase *Bo*Man26B.

Collectively, our results point to a model in which the smaller manno-oligosaccharides generated by *Ri*GH26 are imported through a dedicated β-mannan transport system consisting of *Ri*MnBP/*Ri*MPP1/*Ri*MPP2 (Fig. 7). In the cytoplasm, acetylated and galactosylated manno-oligosaccharides are systematically debranched by *Ri*CE2, *Ri*CEX and *Ri*GH36, and subsequently depolymerized. Removal of glucose units from glucomannan-oligosaccharides is carried out by *Ri*GH3A and *Ri*GH3B. Based on the highest transcriptional and protein regulation, the main depolymerization strategy for breakdown of unsubstituted manno-oligosaccharides is mediated by the activity of two synergistic mannoside phosphorylases (*Ri*GH130_2 and *Ri*GH130_1) and an epimerase (*Ri*Mep), similar to the mannan catabolic pathway proposed in the ruminal bacterium *Ruminococcus albus*[46]. A similar system has been reported in *B. fragilis*[32] and *B. ovatus*[30], although only composed of an epimerase and a mannosylglucose phosphorylase (GH130_1) that, together, process GH26s-generated $M_2$ units. The presence of the manno-oligosaccharide phosphorylases *Ri*GH130_2 allows *R. intestinalis* to process undecorated manno-oligosaccharide of DP > 2, consistent with the internalization of large manno-oligosaccharides generated by *Ri*GH26-hydrolysis of polymeric mannan. However, GH130_2s mainly catalyze the phosphorolysis of undecorated manno-oligosaccharides[47]. Removal of mannose units from substituted substrates is mediated by the reducing end mannose-releasing exo-oligomannosidase *Ri*GH113, which displays a previously undescribed specificity. The two different approaches based on the phosphorylases and the GH113 are likely to be a functional adaptation to accelerate the depolymerization process of mannan. Eventually, mannan catabolism fuels monosaccharide fermentation via glycolysis and leads to the production of butyrate, which is the primary energy source for host colonocytes[5,48]. Colonocytes oxidize butyrate to carbon dioxide[49], thereby keeping the epithelium hypoxic (<1% $O_2$). This condition promotes gut homeostasis by stabilizing the hypoxia-inducible transcription HIF that coordinates barrier protection in the mucosa[50,51]. Recently, it has been shown that antibiotic-mediated depletion of butyrate-producing Clostridia increases colonocytes oxygenation and drives aerobic pathogen expansion in the gut lumen, resulting in *Salmonella enterica*-induced gastroenteritis[52]. Importantly, *R. intestinalis* has been found to affect host histone epigenetic states, direct colonic epithelial cells metabolism away from glycolysis and towards fatty acid metabolism, reduce the levels of inflammatory markers and ameliorate atheriosclerosis in a diet-dependent fashion[18]. The athero-protective effect was in part attributed to butyrate, as this SCFA has been shown to inhibit key inflammatory pathways involved in cardiovascular disease development[18].

The absence of oligosaccharides from *R. intestinalis* AcGGM-spent supernatant (Supplementary Fig. 1a, b) demonstrates that the β-mannan degradation apparatus is optimized for efficient uptake of all the products released by *Ri*GH26, maximize intracellular breakdown and avoid nutrient leakage. This will limit the access to other bacteria, such as *Bacteroides spp*., competing for the same resource. Using AcGGM, we have shown that *R. intestinalis* and *B. ovatus*, which possesses an equally complex β-mannan degrading system, shared the available resources and maintained coexistence (Fig. 6b). Notably, *R. intestinalis* outcompeted *B. ovatus* in the late exponential and stationary phase of growth; these results show that *R. intestinalis* is capable to bind and import the remaining β-mannan breakdown products (preferred by the *Ri*MnBP transport protein) more efficiently than *B. ovatus*. Thus, it is likely that the β-mannan utilization apparatus provides *R. intestinalis* with a selective advantage during nutrient limitation when microbial competition for the available carbohydrates in the gut is intense. Understanding the mechanism by which β-mannan is degraded by key commensal members of the gut is crucial to designing intervention strategies through the use of targeted prebiotics which aim to program or reprogram the composition of the microbiota to maximize human health. Our in vivo study demonstrates that a diet supplemented with AcGGM can be used to manipulate the gut microbiota and to facilitate the growth of species equipped with a β-mannan degrading system, including *R. intestinalis* (Fig. 6e). This is supported by the increase in the relative abundance of *R. intestinalis*, *B. uniformis* and *B. ovatus*, which all possess enzymes able to degrade AcGGM (BACUNI_00371 - BACUNI_00383; BACOVA_02087–02097 and BACOVA_03386–03406 respectively). *R. intestinalis* was highly responsive to the AcGGM within a day, with a 10 to 30 fold increase at the 2.5% and 7.5% AcGGM diet, corroborating its ability to respond dynamically to variation in this dietary fiber. Intriguingly, *R. intestinalis*' response did not last over the 7 day feeding treatment and the acetogen *M. formatexigens* seemed to replace it. A cluster of genes with predicted functions in β-manno-oligosaccharide utilization (BRYFOR_07194- BRYFOR_07206) was identified in the genome of *M. formatexigens* (Supplementary Fig.17a). The results shown in Supplementary Fig. 17b,d suggest that *R. intestinalis* and *M. formatexigens* occupy different metabolic niches in the distal gut; the former consumes complex β-mannans, whereas the acetogen feasts on mono- and oligosaccharides. When in co-culture with either *R. intestinalis* or *B. ovatus*, *M. formatexigens* was outcompeted in vitro (Supplementary Fig. 17e-f). A previous study with gnotobiotic mice bi-associated with the prominent saccharolytic bacterium *B. thetaiotamicron* and *M. formatexigens* showed that the presence of *M. formatexigens* caused a decrease in the cecal levels of *B. thetaiotaomicron*, compared with mono-associated controls. Transcriptional and metabolic analyses demonstrated that *M. formatexigens* is capable of consuming a variety of plant-derived oligosaccharides and microbial and host-derived *N*-glycans (such as *N*-acetylglucosamine), suggesting that this ability could confer a fitness advantage when competing with the glycan-consuming *Bacteroides*[53]. Thus, it is likely that, when present as part of the synthetic microbial community described in this paper, *M. formatexigens* may be indirectly benefiting of either manno-oligosaccharides feeding/cross-feeding with other microorganisms or by its ability to grow mixotrophically, simultaneously utilizing organic carbon sources and formate or $H_2$ for energy[53]. Notably, *M. formatexigens* outcompeted *B. ovatus* at the 7.5% AcGGM diet, underscoring the competitiveness of this acetogen in a community setting. In the context of a complex microbial community, it is likely that *M. formatexigens* makes an important contribution to host nutrition improving fermentation by acting as a formate or $H_2$ sink and by generating acetate as main metabolic product[53].

Collectively, diet-induced changes involved the promotion of mannanolytic bacteria producing propionate, acetate and butyrate, metabolites that are known to regulate hepatic lipid, glucose homeostasis and health of the intestinal hepithelium[11]. These SCFA-producers gained a competitive advantage over colonic mucin-degrading bacteria. Given that intermittent dietary fiber-deprivation results in a thinner mucus layer in mice, eventually

enhancing pathogen susceptibility[42], our results support the concept that β-mannan-based interventions not only could contribute to preventing mucus barrier dysfunctions but also maintaining a gut environment that keeps pathogenic bacteria away. If confirmed in humans, these findings may help to prevent diseases affecting the integrity of the colonic mucus layer, such as ulcerative colitis[54]. Indeed, the fact that the β-mannan degradation pathway is a core trait found in the majority of the human gut microbiota[55] highlights the relevance of potential therapeutic interventions through the use of β-mannan formulations to the general population.

## Methods

**Glycans**. Carbohydrate substrates used in this study are listed in Supplementary Table 6. All glycan stocks were prepared at 10 mg ml$^{-1}$ in ddH$_2$O and sterilized by filtration using a 0.22 μm membrane filter (Sarstedt AG & Co, Germany).

**Bacterial strains and growth conditions**. Unless otherwise stated, *R. intestinalis* L1–82[23] was routinely grown at 37 °C without agitation in an anaerobic cabinet (Whitley A95 workstation, Don Whitley, UK) under an 85% N$_2$/10% H$_2$/ 5% CO$_2$ atmosphere. Growth experiments were carried out in YCFA medium (YCFA—Yeast extract-Casein hydrolysate-Fatty Acids)[56] supplemented with 0.5% (w/v) of the specific carbohydrate to be examined. Overnight cultures (300 μl) were used to inoculate 30 ml aliquots of YCFA plus the carbohydrate to be tested. These pre-cultures were passaged at least three times on the same media to ensure cell growth adaptation on a single carbon source prior to inoculation of the final cultures for growth experiments, RNA-sequencing and proteomic analysis. Bacterial growth was determined spectrophotochemically by monitoring changes in the optical density at 600 nm (OD$_{600}$). In addition, growth on turbid substrates was assessed by measuring differences in pH compared to that of starting medium. Growth and pH curves are averages of three biological replicates, with two technical replicates each. Routine culturing of *Bacteroides ovatus* ATCC 8483 and *M. formatexigens* DSM 14469 was in anaerobic Chopped Meat Medium[57] under static conditions at 37 °C.

**Transcriptomic analysis by RNA sequencing**. *R. intestinalis* was cultured in triplicate on YCFA supplemented with 0.5% (w/v) glucose, galactose, KGM or AcGGM as described above. Cells were harvested at mid-exponential phase and RNA was extracted using the RNeasy Mini Kit (Quiagen) according to the manufacturer's instructions. RNA-seq libraries were prepared using the ScriptSeq Complete kit from Epicentre. Samples were paired-end sequenced on an Illumina Hiseq 4000 instrument at Beijing Genomics Institute (BGI). Analysis of the RNA-seq results was performed exactly as described in[25]. Differential gene expression analysis was performed with the DeSeq2 package[58].

**Cloning, overexpression, and protein purification**. The genes encoding mature forms of the proteins described in this study were amplified from the *R. intestinalis* L1–82 genomic DNA (BioProject accession number PRJNA30005 [https://www.ncbi.nlm.nih.gov/bioproject/PRJNA30005]) by PCR, using appropriate primers (Supplementary Table 7). PCR products were generated using the Q5 High-Fidelity DNA Polymerase (New England BioLabs, United Kingdom) with 50 ng DNA as template. Prior to cloning the DNA fragment encoding *Ri*GH1_D2 (ROSINTL182_05469), sequence ambiguities at the 3'-end of ROSINTL182_05470 were corrected through sequencing the PCR product generated with the primers listed in the Supplementary Table 8. The gene ROSINTL182_07683 was synthesized without the N-terminal signal sequence predicted by SignalP v.4.1[59] (residues 1–27 from transcription start). The PCR amplicons were cloned into the pNIC-CH expression vector by ligation-independent cloning (LIC)[60]. The gene encoding *Ri*Mnbp (ROSINTL182_05479) was cloned in the vector pETM-11 following the method described elsewhere[25]. Recombinant proteins generally contained a C-terminal His$_6$-tag, although, in some cases, His-tag translation was prevented by the introduction of one or two stop codons at the end of the open-reading frame (*Ri*Mep, *Ri*GH36, *Ri*Pgm and *Ri*GH113). The His$_6$-tag was excluded to prevent interaction with putative C-terminal active or catalytic residues that could be detrimental to the enzymes' activity. Constructs were verified by sequencing (Eurofins, UK). Proteins were produced in *E. coli* BL21 Star (DE3) cells (Invitrogen) as previously described[61]. Briefly, cells were cultured to mid-exponential phase in Tryptone Yeast extract (TYG) containing 50 mg ml$^{-1}$ kanamycin at 25 °C. Protein overexpression was induced by adding isopropyl β-D-thiogalactopyranoside (IPTG) to a final concentration of 200 μM, followed by incubation for a further 16 h at 25 °C. Cells were harvested by centrifugation, sonicated and recombinant proteins were purified by either immobilized metal ion affinity chromatography (IMAC) or hydrophobic interaction chromatography (HIC). For IMAC purification, the clarified cell lysate was loaded onto 5 ml HisTrap HP Ni Sepharose columns (GE Healthcare) connected to an ÄKTA purifier FPLC system (GE Healthcare). Protein elution was achieved by using a

linear gradient from 5 to 500 mM imidazole. *Ri*GH113, *Ri*GH36, *Ri*Mep and *Ri*Pgm were purified by HIC by loading the cell-free broth, adjusted to buffer A (1.5 M ammonium sulfate), onto a 5 ml HiTrap Phenyl FF (GE Healthcare) equilibrated with the same buffer. Protein was eluted by using a linear reverse gradient to 100 mM NaCl over 90 min at a flow rate of 2.5 ml min$^{-1}$. After IMAC and HIC, samples were concentrated and further purified by size exclusion chromatography (SEC) using a HiLoad 16/60 Superdex G75 size exclusion column (GE Healthcare) and a running buffer consisting of 20 mM Tris-HCl pH 8.0 with 200 mM NaCl. Fractions containing the pure protein were combined, concentrated and buffer exchanged to 20 mM Tris pH 8.0, using a Vivaspin 20 (10-kDa molecular weight cutoff) centrifugal concentrators (Sartorius Stedim Biotech GmbH, Germany). Protein purity was estimated to be over 95% for all the enzymes using SDS-PAGE. Protein concentrations were determined using the Bradford assay (Bio-Rad, Germany).

**Glycoside hydrolase and phosphorylase activity assays**. Enzyme assays, unless otherwise stated, were carried out in 10 mM sodium phosphate buffer, pH 5.8, for up to 16 h at 37 °C and 700 rpm. Reactions with *Ri*GH130_1 and *Ri*GH130_2 were prepared in 100 mM sodium phosphate buffer, pH 5.8. The activity of *Ri*Pgm against M1P and G1P was tested in 10 mM sodium phosphate buffer, pH 5.8, supplemented with 1 mM MgCl$_2$. To determine the specificity of *Ri*GH113, the recombinant protein was sequentially incubated with 0.1 mg ml$^{-1}$ pre- reduced or oxidized manno-oligosaccharides at 37 °C overnight. Reduction of manno-oligosaccharides was conducted by incubating 1 mg ml$^{-1}$ manno-oligosaccharides in a volume of 75 μl with sodium borodeuteride (NaBD$_4$; 0.5 M in 100 mM NaOH) solution. The reaction was incubated overnight at room temperature then quenched by adding 25 μl of 25 mM sodium acetate. Oxidation of manno-oligosaccharides reducing-end was obtained by incubating the substrates (1 mg ml$^{-1}$) with the *Neurospora crassa* cellobiose dehydrogenase (*Nc*CDH) overnight at 37 °C. Both NaBD$_4$ and *Nc*CDH pretreated samples were diluted 10X in standard assay buffer before addition of *Ri*GH113. Between three and five independent experiments were performed to determine the enzyme activities.

**MALDI-TOF mass spectrometry of reaction products**. Reaction products generated by the enzymes used in this study were analyzed by matrix-assisted laser desorption/ionization time of flight mass spectrometry (MALDI-TOF MS) as described previously[62]. Briefly, 2 μl of a matrix, consisting of 9% 2,5-dihydroxybenzoic acid (DHB) in 30% acetonitrile, were applied to an MTP 384 ground steel target plate TF (Bruker Daltonics, Germany). Sample (1 μl) was then mixed with the matrix and dried under a stream of warm air. Samples were analyzed with an Ultraflex MALDI-ToF/ToF instrument (Bruker Daltonics, Germany), equipped with a Nitrogen 337 nm laser beam and operated in positive acquisition mode. Results were analyzed using the Bruker FlexAnalysis software (version 3.3).

**HPAEC-PAD**. Mono- and oligosaccharides products were analyzed on a Dionex ICS-3000 HPAEC system operated by the Chromeleon software version 7 (Dionex, Thermo Scientific), as described previously[62]. Sugars were injected onto a Carbo-Pac PA1 2 × 250-mm analytical column (Dionex, Thermo Scientific) coupled to a CarboPac PA1 2 × 50-mm guard column kept at 30 °C. Manno-oligosaccharides and phosphorylated monosaccharides were eluted in 0.1 M NaOH at a flow rate of 0.25 ml min$^{-1}$ by increasing the concentration of sodium acetate (NaOAc) exponentially from 0 to 0.3 M over 26 min (from 9 to 35 min after injection), before column reconditioning by 0.1 M NaOH for 10 min. Commercial manno-oligosaccharides with DP2−6 were used as standards. For cello-oligosaccharides, the separation was done using a multistep linear gradient going from 0.1 M NaOH to 0.1 M NaOH–0.1 M NaOAc over 10 min, 0.1 M NaOH–0.14 M NaOAc after 14 min, 0.1 M NaOH–0.3 M NaOAc at 16 min followed by a 2 min exponential gradient to 1 M NaOAc, before reconditioning with 0.1 M NaOH for 9 min. Cello–oligosaccharides with DP 2−6 were used as standards. For the analysis of disaccharides (G$_1$M$_1$ or M$_1$G$_1$) and phosphorylated monosaccharides generated from the activity of *Ri*GH130_2, *Ri*Mep, *Ri*GH130_1, *Ri*Pgm and *Ri*GH1, the elution was done at 0.25 ml min$^{-1}$ using a 40 min program. The program started with 0.01 M potassium hydroxide (KOH) for 15 min, reaching the concentration of 0.1 M KOH at 25 min after injection and was kept for additional 5 min at the same KOH concentration. Between each sample, the column was re-equilibrated by running initial conditions for 10 min.

**Protein cellular localization**. Proteins of interest were detected using anti-sera raised in rats (Eurogentec) against the corresponding recombinant *Ri*GH26 or the previously characterized RiXyn10A[25].

For immunofluorescence microscopy, *R. intestinalis* cells were grown in YCFA containing 0.5% AcGGM, wheat arabinoxylan (WAX) or glucose to an OD$_{600}$ of 0.8, collected by centrifugation (4000 × *g* for 5 min) and washed twice in phosphate buffered saline (PBS). Cells were resuspended in 500 μl PBS and fixed by adding an equal volume of 2 × formalin (9% formaldehyde in PBS) on ice for 30 min. The bacterial pellet was washed twice with 1 ml PBS prior to resuspension in 1 ml of blocking buffer (1% bovine serum albumin, BSA, in PBS) and incubation at 4 °C for 16 h. After incubation the cell pellets were harvested by centrifugation and the supernatant discarded. For labelling, the bacteria were incubated with 0.5 ml of anti-sera (diluted 1:500 in blocking buffer) for 2 h at 25 °C. The cells were then

pelleted, washed with 1 ml PBS and resuspended in 0.5 ml goat anti-rat IgG Alexa-Fluor 488 (Sigma-Aldrich), diluted 1:500 in blocking buffer and incubated 1 h at 25 °C. The cells were again harvested, washed with 1 ml PBS and suspended in 100 μl PBS containing one drop of ProLong Gold antifade reagent (Life Technologies). Labeled bacterial cells were mounted onto glass slides and secured with coverslips. Fluorescence microscopy was performed on a Zeiss AxioObserver equipped with the ZEN Blue software. Images were acquired using an ORCA-Flash4.0 V2 Digital CMOS camera (Hamamatsu Photonics) through a 100x phase-contrast objective. A HXP 120 Illuminator (Zeiss) was used as a fluorescence light source.

**Analysis of the bacterial proteome.** *R. intestinalis* was grown in triplicate on YCFA supplemented with either 0.5% (w/v) glucose or AcGGM, respectively, as a sole carbon source. Samples (10 ml) were harvested at the mid-exponential growth phase. Cell pellet was collected by centrifugation (4500 × *g*, 10 min, 4 °C), resuspended in 50 mM Tris-HCl, 0.1% (v/v) Triton X-100, 200 mM NaCl, 1 mM dithiothreitol and disrupted by bead-beating using three 60 s cycles with a FastPrep24 (MP Biomedicals, CA). Proteins were precipitated with ice-cold trichloroacetic acid (TCA), final concentration of 10% (v/v), incubated on ice for 1 h, centrifuged (15,000 × *g*, 15 min, 4 °C) to pellet the precipitated proteins and washed with 300 μl ice-cold 0.01 M HCl in 90% acetone. Proteins were separated by SDS-PAGE with a 10% Mini-PROTEAN gel (Bio-Rad Laboratories, CA) and then stained with Coomassie brilliant blue R250. The gel was cut into five slices, after which proteins were reduced, alkylated, and in-gel digested according to a method published previously[63]. The peptides were dried under vacuum, solubilized in 0.1% (v/v) trifluoroacetic acid (TFA) and desalted using $C_{18}$ ZipTips (Merck Millipore, Germany) according to the manufacturer's instructions.

The peptide mixture from each fraction was analyzed using a nanoHPLC-MS/MS system as described previously[63], using a Q-Exactive hybrid quadrupole-orbitrap mass spectrometer (Thermo Scientific) equipped with a nano-electrospray ion source. Mass spectral data were acquired using Xcalibur (v.2.2 SP1).

MS raw files were processed with the MaxQuant software suite[64] (version 1.4.1.2) for identification and label-free quantification (LFQ) of proteins. Proteins were identified by searching MS and MS/MS data of peptides against the UniProtKB complete proteome of *R. intestinalis* L1–82 (4698 sequences) supplemented with common contaminants (e.g., keratins, trypsin, and bovine serum albumin). In addition, reversed sequences of all protein entries were concatenated to the database for estimation of false-discovery rates (FDRs). Trypsin was set as proteolytic enzyme and two missed cleavages were allowed. Protein N-terminal acetylation, oxidation of methionines, deamidation of asparagines and glutamines and formation of pyro-glutamic acid at N-terminal glutamines were defined as variable modifications while carbamidomethylation of cysteines was used as a fixed modification. The "match between runs" feature of MaxQuant, which enables identification transfer only between samples from the same carbon source based on accurate mass and retention time, was applied with default parameters. All identifications were filtered in order to achieve a protein FDR of 1%. A protein was considered "present" if it was detected in at least two of the three biological replicates in at least one glycan substrate. Missing values were imputed from a normal distribution (width of 0.3 and down shifted 1.8 standard deviations from the original distribution) in total matrix mode and differential abundance analysis was performed using an unpaired two-tailed Student's *t*-test with a permutation-based FDR set to 0.05. Hierarchical clustering and heat map representations were generated using Euclidean distance measure and average linkage using Perseus[65] (version 1.5.5.3).

**Substrate binding assay using SPR.** The affinity of *Ri*CBM27 and *Ri*CBM23 to soluble manno-oligosaccharides and cello-oligosaccharides was evaluated by SPR using a Biacore T100 (GE Healthcare). The two CBMs, diluted to 10 mM sodium acetate (pH 4.1) to 2.3 μM, were immobilized on a NTA sensor chip (GE Healthcare) to a density of 3000−4000 response units (RU). Sensograms were recorded at 25 °C in phosphate/citrate buffer (20 mM phosphate/citrate buffer; 150 mM NaCl; pH 6.5, 0.005% (v/v) P20 surfactant) at 30 μl per min with association and dissociation times of 90 s and 240 s, respectively. CBMs binding was tested towards 0.2 nM – 1 mM of carbohydrate ligands dissolved in the same buffer as above. Data were analyzed using the Biacore T100 evaluation software, and equilibrium dissociation constants ($K_d$) were obtained by fitting a single-site binding model to either the steady-state response data or the full sensograms.

**ITC.** Binding of manno-oligosaccharides to *Ri*MnBP was measured at 25 °C in 10 mM sodium phosphate pH 6.5 using an $ITC_{200}$ microcalorimeter (MicroCal). *Ri*MnBP in the sample cell was titrated by 19 injections of carbohydrate ligand separated by 120 s. The following concentrations were used: 900 μM of $M_3$ in the syringe and 76.5 μM *Ri*MnBP in the sample cell; 1365 μM of $M_4$ or $M_5$ in the syringe and 91 μM *Ri*MnBP in the sample cell; 2270 μM of $M_6$ in the syringe and 117 μM *Ri*MnBP in the sample cell; 750 μM of diacetylated mannotetraose ($M_4Ac_2$) in the syringe and 50 μM *Ri*MnBP in the cell; 1500 μM of diacetylated manno-pentaose ($M_5Ac_2$) in the syringe and 100 μM *Ri*MnBP in the cell. Thermodynamic binding parameters were determined using the MicroCal Origin software (version 7.0).

**Competition experiments.** *R. intestinalis*, *B. ovatus* and *M. formatexigens* cells were grown overnight under anaerobic conditions in YCFA containing 0.5% (w/v) AcGGM (YCFA-AcGGM) as the sole carbon source. These subcultures were used to inoculate, in approximately equal proportions (estimated by OD_600), 30 ml of the same medium. A control culture of YCFA-AcGGM was also inoculated with either *R. intestinalis*, *B. ovatus* or *M. formatexigens*. Growth (OD_600) was monitored for up to 24 h, withdrawing 1 ml samples for quantitative PCR (qPCR) analysis at selected time points. Cells were pelleted, combined with 200 μl of TE buffer (pH 7.8) and bead-beated for 2 min (FastPrep96, MP Biomedicals, CA) using ≤ 106 μm acid-washed glass beads (Sigma-Aldrich). Genomic DNA was extracted using the Mag Midi kit (LGC Group, UK) according to the manufacturer's instructions. qPCR was performed in a LightCycler 480 II system (Roche, Germany) using specific primers for each strain (Supplementary Table 9). In addition, a high-resolution melting (HRM) analysis was conducted to evaluate the specificity of the amplification and the lack of primer dimers. The raw data were imported into the LinReg PCR program[66] and the calculated Cq values and PCR efficiency were used to deduce the ratio of *R. intestinalis*, *B. ovatus* and *M. formatexigens* at each time point. Statistically significant differences were determined using the unpaired two-tailed Student's *t*-test.

**Human gut microbiota-associated mice and diets.** All experiments involving animals complied with all relevant ethical regulations for animal testing and research and were approved by the University of Michigan, University Committee for the Use and Care of Animals. Germfree mice were colonized with a synthetic microbiota composed of 14 fully sequenced human species according to the methodology previously adopted by Desai et al[42]. Briefly, seven 6-week-old germfree male wild-type Swiss Webster mice that had been raised on *ad libitum* access to a high fiber chow diet (LabDiet 5013) and autoclaved distilled water were gavaged for 3 consecutive days with 200 μl each day of a mixture of the 14 different species. Colonized mice were maintained on this high fiber diet for 14 days before being switched to a series of diet regimes with varying fibers. This feeding sequence consisted of 7 days of feeding on a gamma-irradiated fiber-free (FF) diet (TD.140343, Harlan Teklad, USA) that does not contain AcGGM or related molecules. Mice were then switched for 7 days to a custom version of the same diet that contained AcGGM at 2.5% w/w, followed by a 7-day washout period on the FF diet, and finally 7 days of feeding on a version of this diet containing AcGGM at 7.5% w/w (in both AcGGM diets an equivalent amount of glucose was removed to accommodate the prebiotic addition). Fecal samples were taken 1 day before and 1 day after each diet transition, effectively allowing us to measure changes in response to AcGGM supplementation at 1 and 7 days post exposure relative to the FF diet. The relative abundance of each microbial strain at sampled time points was measured by qPCR, using previously validated primer sets, from total DNA extracted from freshly voided fecal pellets (stored at −20 °C until extraction) exactly as described previously[42]. Statistically significant differences were determined using the unpaired two-tailed Student's *t*-test.

**Comparative genomic analysis.** Identification of similar β-mannan catabolic genes in bacteria belonging to the Clostridium XIVa cluster was performed using the Gene Ortholog Neighborhood viewer on the Integrated Microbial Genomes website (https://img.jgi.doe.gov). This was done using the genes encoding *Ri*GH26 (ROSINTL182_07683, GenBank ABYJ02000124.1:7167–11129) and *Ri*MEP (ROSINTL182_05476, GenBank ABYJ02000025.1:3200–4429) as the search homolog and the default threshold e-value of 1e-5. Then, a sequence comparison was conducted where each *R. intestinalis* L1–82 RefSeq annotated protein sequence was subjected to BLASTp searches against other Clostridium XIVa members. Sequences with coverage <60% and amino acid similarity <45% were excluded.

**Reporting summary.** Further information on experimental design is available in the Nature Research Reporting Summary linked to this article.

## Data availability

All data supporting the findings of this study are available within the article and Supplementary Information, or from the corresponding author upon request. The transcriptomic data described in this article are submitted under NCBI BioProject accession number PRJNA516396. The mass spectrometry proteomics data have been deposited to the ProteomeXchange Consortium via the PRIDE partner repository with the dataset identifier PXD012448.

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

## Acknowledgements

The authors wish to thank M. Kjos (NMBU) for assistance with immunofluorescence microscopy. This work was supported by a grant from the Research Council of Norway (244259). Additional funding was from a Graduate School DTU Scholarship, Lyngby, Denmark for M.L.L. and from the Independent Research Fund Denmark, Natural Sciences (DFF, FNU) by a Research Project 2 grant (Grant ID: 4002–00297B) to M.A.H. Carlsberg Foundation is also acknowledged for an instrument grant for purchase of the ITC200 calorimeter to M.A.H.

## Author contributions

Experiments were primarily designed by S.L.L.R., E.C.M., M.A.H., and B.W. S.L.L.R. cloned, expressed, purified and performed functional characterizations of the enzymes. Production of AcGGM was performed by L.M. and B.W. The initial growth experiments on mannans were performed by M.L.L. and these experiments were used to prepare RNA and performed the transcriptional analysis together with C.T.W. Proteomic analysis was done by S.L.L.R. and M.Ø.A. SPR and ITC was performed by M.E.H., who also cloned, expressed and purified the transport protein. S.L.L.R. and G.P. conducted the in vitro growth experiments, competition experiments and qPCR. S.L.L.R. performed enzyme localization studies. Mice experiment was conducted by N.A.P., R.G. and E.C.M. The manuscript was written primarily by S.L.L.R. and B.W. with contributions from P.B.P., M.Ø.A., M.A.H., and E.C.M. Figures were prepared by S.L.L.R., M.E.H., and N.A.P. All authors reviewed the final manuscript.

## Additional information

**Competing interests:** The authors declare no competing interests.

