## [Peer Review File · Nature Communications]

Reviewers' comments:

Reviewer #1 (Remarks to the Author):

Manuscript NCOMMS-18-21943-T

The authors have demonstrated through a combination of multi “omic” analyses and biochemical studies that *R. intestinalis* can break down plant cell wall polysaccharides β -Mannans, a common component of the human diet. Their detailed biochemical studies of the encoded enzymes have resulted in a model of sequential action for the mannan utilization system encoded by *R. intestinalis*. Additionally, their in vivo experiment suggests that the use of β -mannan-based prebiotics could promote SCFA-producers and provide a competitive advantage over colonic mucin-degrading bacteria.

These findings are novel, the work is thorough and convincing, and the paper should be of interest to a wide audience.

I was somewhat surprised to see that *R. intestinalis* relative abundance is only significantly increased on the first day of AcGGM treatment and returns to near baseline by day 7. *M. formatexigens* must be very efficient once it had a start on the AcGGM to then be able to outcompete/suppress *B. ovatus* and *R. intestinalis* by the end of the week trial. Perhaps the authors could consider competition experiments with this acetogen as well.

Minor editorial comments:

Line 43: 'by the human' should be 'by human'

Line 52: bifidobacteria and lactobacilli to be written in lower case, and 'strains' can be removed (and perhaps it is better to say: 'certain bifidobacterial and lactobacilli')

Line 54: typo: 'Fimicutes' should be 'Firmicutes'

Line 62: typo: 'colonize' should be 'colonizes'

Line 80: typo: 'shows' should be 'show'

Line 85: typo: 'alter' should be 'alters'

Line 101: 'deriving' should be 'derived'

Line 126: probably better to replace 'needed' by 'required'

Line 148: either 'predicted' or 'putative' can be deleted

Lines 161-162: the sentence '.....importer were amongst etc' is more clear/accurate if written as follows '....importer were shown to exhibit the highest level of increased expression during growth on beta-mannan (and when compared to growth on glucose).

Line 189: 'resistant for' should be 'resistant to'

Lines 277-278: 'The recombinant etc'; this statement does not mean anything without providing some information on what conditions and substrates were tested.

Lines 282-296: the experiments described were performed in batch cultures without pH control and the observed findings could have been skewed by differences in intrinsic acid sensitivities between the two strains. Perhaps the authors may mention this.

Line 302: 'with varying' should be 'with a varying'

Line 310: 'amino acids' should be 'amino acid'

Line 340: 'may suggest' should be 'suggests'

Line 341: 'against' should be 'with'

Reviewer #2 (Remarks to the Author):

The manuscript by La Rosa et al is concerned with the breakdown of β -mannan by *Roseburia intestinalis*. They show that two loci are expressed in response to β -mannan and they describe detailed activities for the enzymes involved in β -mannan degradation. The authors describe the endo-mannanase activity of RiGH26 and hypothesise on the role of two CBM modules to anchor manno-oligosaccharides to the surface of the cell. The authors also report a new specificity for RiGH113 as a reducing end mannose-releasing exo-oligomannosidase. The authors also describe a number of enzyme activities involved in β -mannan degradation and go on to predict a model for β -mannan degradation by *R. intestinalis*. The authors also test the ability of *R. intestinalis* to compete with another β -mannan degrader *Bacteroides ovatus* showing that the two can co-exist whilst using β -mannan as a carbon source. Notably, they observed *R. intestinalis* was able to out compete *B. ovatus* during stationary phase suggesting *R. intestinalis* has a selective advantage during nutrient limitation. Finally the authors showed that in a minimal microbiota grown on β -mannan members capable of breaking down β -mannan are enriched and that mucin degrader levels decreased.

This paper is of significant interest to the field as it details the degradation of complex polysaccharide by the firmicute *R. intestinalis*, most studies of polysaccharide degradation in the human gut have centred on the bacteroides. It provides new insights and adds depth to the body of knowledge which could influence the development of targeted prebiotics. The paper has a significant amount of data including bacterial growths, regulation data, enzyme activities, competition experiments and in vivo experiments. It is clear to see that the paper is a considerable amount of work by the authors.

Major points to address.

Fig.2a The authors show growth curves for *R. intestinalis* on different carbon sources. The figure shown only displays growth curves over 8hrs which only covers growth into exponential phase. Usually, I would expect to see a full growth curve right through to stationary phase. Later in the manuscript the authors show HPLC traces of culture supernatant from stationary phase growths. It would be beneficial to show the full growth curve in Fig2a.

Supplementary Fig 1.a. The chromatogram is shown from 2.5 min. There appears to be peaks that would align with the mannose and oligo standards. The Authors state that there was no observed build-up of oligos or mannose, what are the peaks between 2.5 and 10 min?

Page 10 line 201 onwards, the authors claim that the frequency of galactose substitutions does not affect RiGH36 catalytic efficiency. From the data shown it is hard to come to that conclusion, the authors show end point assays displaying RiGH36 is capable of fully hydrolysing oligos with one or two internal substitutions, this does not show how quickly or efficiently the oligos are hydrolysed. To assess the catalytic activity the authors need to either measure the catalytic parameters of the enzyme for each substrate or a time course showing the oligos are degraded in a similar fashion/speed using the same conditions for both oligos.

Finally the authors show a model of β - mannan degradation based on the assumption RiGH26 is located on the cell surface and that is the key enzyme which breaks down β - mannan to oligos for transport by the ABC transporter. I think if this is the case the authors need to show the cellular location for RiGH26 experimentally.

Minor points.

The supplemental figures need to be re-ordered so that they appear in the order they are cited in the text.

Also there is no consistency in how the HPLC traces are displayed and labelled. Some have different colours, some are labelled beneath the trace or above. It would be helpful if the authors have a uniform way to display the HPLC traces. Also, some of the traces were hard to see as the plots are too close together and should be spaced out more. For example Supplemental fig 7.b it is very difficult to see the gal peak as it overlays with another plot and they both have a similar colour.

Subject: Response to reviewers' comments of the manuscript NCOMMS-18-21943-T entitled "The Human Gut Firmicute *Roseburia intestinalis* is a Primary Degradator of Dietary β -Mannans".

We thank the editor and the reviewers for the valuable comments to our manuscript. We now submit a revised version of NCOMMS-18-21943-T that addresses their concerns. Our point-by-point answers to the comments are dealt with chronologically. Our responses are in blue lettering.

Reviewer #1

The authors have demonstrated through a combination of multi "omic" analyses and biochemical studies that *R. intestinalis* can break down plant cell wall polysaccharides β -Mannans, a common component of the human diet. Their detailed biochemical studies of the encoded enzymes have resulted in a model of sequential action for the mannan utilization system encoded by *R. intestinalis*. Additionally, their *in vivo* experiment suggests that the use of β -mannan-based prebiotics could promote SCFA-producers and provide a competitive advantage over colonic mucin-degrading bacteria.

These findings are novel, the work is thorough and convincing, and the paper should be of interest to a wide audience.

We appreciate the reviewer's interest in our data, his/her recognition that the work provides novel insights into poorly characterized β -mannan utilization in Firmicutes and the comments to further improve the manuscript.

- 1) I was somewhat surprised to see that *R. intestinalis* relative abundance is only significantly increased on the first day of AcGGM treatment and returns to near baseline by day 7. *M. formatexigens* must be very efficient once it had a start on the AcGGM to then be able to outcompete/suppress *B. ovatus* and *R. intestinalis* by the end of the week trial. Perhaps the authors could consider competition experiments with this acetogen as well.

Re: Although we agree with the reviewer's comment that would be of interest to conduct *in vivo* competition experiments to show that *M. formatexigens* is able to outcompete *B. ovatus* and *R. intestinalis*, we are currently unable of

performing additional murine experiments. Unfortunately, setting up a new bi-associated mice feeding trial is not feasible due to very limited availability of gnotobiotic mice in Prof. Martens lab, costs and time constrains. Perhaps most importantly, we are truly reluctant to try to justify sacrificing more animals for this purpose. Due to the above points, we have conducted *in vitro* experiments and results are shown in the Supplementary Fig. 17. Overall, our findings show that *M. formatexigens* possesses a degradation apparatus for β -manno-oligosaccharides (Supplementary Fig.17a), is able to grow (Supplementary Fig.17b) and utilizes the manno-oligosaccharides contained in the AcGGM preparation (Supplementary Fig.17c-d) *in vitro*. However, we were surprised to find that *M. formatexigens* did not outcompete either *B. ovatus* or *R. intestinalis* in an *in vitro* setting (Supplementary Fig.17e-f). Taken together with the *in vivo* experiment results shown in Fig. 6e, it is likely that, when present as part of a microbial community, *M. formatexigens* may be indirectly benefiting by either feeding/cross-feeding on manno-oligosaccharides or another mechanism. Even though we do not know the biological significance of this finding without additional experimental evidence which is outside the scope of this manuscript, we have added the following hypotheses to the discussion lines 419-439:

“Intriguingly, *R. intestinalis*’ response did not last over the 7 day feeding treatment and the acetogen *M. formatexigens* seemed to replace it. A cluster of genes with predicted functions in β -manno-oligosaccharide utilization (BRYFOR_07194- BRYFOR_07206) was identified in the genome of *M. formatexigens* (Supplementary Fig.17a). The results shown in Supplementary Fig. 17b-d suggest that *R. intestinalis* and *M. formatexigens* occupy different metabolic niches in the distal gut; the former consumes complex β -mannans, whereas the acetogen feasts on mono- and oligosaccharides. When in co-culture with either *R. intestinalis* or *B. ovatus*, *M. formatexigens* was outcompeted *in vitro* (Supplementary Fig. 17e-f). A previous study with gnotobiotic mice bi-associated with the prominent saccharolytic bacterium *B. thetaiotamicron* and *M. formatexigens* showed that the presence of *M. formatexigens* caused a decrease in the cecal levels of *B. thetaiotaomicron*, compared with mono-associated controls. Transcriptional and metabolic

analyses demonstrated that *M. formatexigens* is capable of consuming a variety of plant-derived oligosaccharides and microbial and host-derived *N*-glycans (such as *N*-acetylglucosamine), suggesting that this ability could confer a fitness advantage when competing with the glycan-consuming *Bacteroides* (Rey et al., 2010 JBC). Thus, it is likely that, when present as part of the synthetic microbial community described in this paper, *M. formatexigens* may be indirectly benefiting by either manno-oligosaccharides feeding/cross-feeding with other microorganisms or by its ability to grow mixotrophically, simultaneously utilizing organic carbon sources and formate or H₂ for energy (Rey et al., 2010 JBC)".

And the following text to the discussion lines 441-444:

"In the context of a complex microbial community, it is likely that *M. formatexigens* makes an important contribution to host nutrition improving fermentation by acting as a formate or H₂ sink and by generating acetate as main metabolic product (Rey et al., 2010 JBC)".

Minor editorial comments:

Re: We thank the reviewer for detecting these; corrections have been made in the text.

- 1) Line 43: 'by the human' should be 'by human'.
Re: Done as suggested.
- 2) Line 52: bifidobacteria and lactobacilli to be written in lower case, and 'strains' can be removed (and perhaps it is better to say: 'certain bifidobacterial and lactobacilli').
Re: Done as suggested.
- 3) Line 54: typo: 'Fimicutes' should be 'Firmicutes'.
Re: Done as suggested.
- 4) Line 62: typo: 'colonize' should be 'colonizes'.
Re: Done as suggested.
- 5) Line 80: typo: 'shows' should be 'show'.
Re: Done as suggested.
- 6) Line 85: typo: 'alter' should be 'alters'.

Re: Done as suggested.

- 7) Line 101: 'deriving' should be 'derived'.

Re: Done as suggested.

- 8) Line 126: probably better to replace 'needed' by 'required'.

Re: Done as suggested.

- 9) Line 148: either 'predicted' or 'putative' can be deleted.

Re: Done as suggested.

- 10) Lines 161-162: the sentence '.....importer were amongst etc' is more clear/accurate if written as follows '....importer were shown to exhibit the highest level of increased expression during growth on beta-mannan (and when compared to growth on glucose).

Re: Done as suggested.

- 11) Line 189: 'resistant for' should be 'resistant to'.

Re: Done as suggested.

- 12) Lines 277-278: 'The recombinant etc'; this statement does not mean anything without providing some information on what conditions and substrates were tested.

Re: As per reviewer's suggestion, we have now added this information at line 283-286.

- 13) Lines 282-296: the experiments described were performed in batch cultures without pH control and the observed findings could have been skewed by differences in intrinsic acid sensitivities between the two strains. Perhaps the authors may mention this.

Re: We agree with the reviewer that pH is an important factor influencing the competition between bacterial species and the response of human gut (fecal) communities to dietary fibers (Chung *et al.* BMC Biology, 2016).

The growth experiments with a simplified microbial community showed in Fig. 6a-d were performed in a buffered system and the pH of the stationary phase cultures after growth on AcGGM was 5.8 ± 0.16 . If the findings of the competition experiments on AcGGM were the results of differences in acid sensitivities between the two strains, we believe that we should have observed the same outcome when *R. intestinalis* and *B. ovatus* were co-cultivated on mannose, where the pH of the stationary phase culture was 5.6 ± 0.11 . Additionally, using the same culturing conditions, Leth *et al.* has

shown that *R. intestinalis* is able to outcompete *B. ovatus* during the exponential-phase of growth on xylotetraose and xylans (Leth et al., Nature Microbiol, 2018), whereas the proportional representation of *B. ovatus* increased in the stationary phase. Due to the above points, we hope that the reviewer will agree with us that the findings of the *in vitro* experiment are not the result of a pH-effect.

We have now added this information in the legend of Fig. 6 lines 1003-1005: “The pH of the stationary phase cultures after growth on either AcGGM or mannose was 5.8 ± 0.16 and 5.6 ± 0.11 , respectively, thus showing that the results are not due to differences in acid sensitivity between the two strains”.

14) Line 302: ‘with varying’ should be ‘with a varying’.

Re: Done as requested.

15) Line 310: ‘amino acids’ should be ‘amino acid’.

Re: Done as requested.

16) Line 340: ‘may suggest’ should be ‘suggests’.

Re: Done as requested.

17) Line 341: ‘against’ should be ‘with’.

Re: Done as requested.

Reviewer #2

The manuscript by La Rosa *et al* is concerned with the breakdown of β -mannan by *Roseburia intestinalis*. They show that two loci are expressed in response to β -mannan and they describe detailed activities for the enzymes involved in β -mannan degradation. The authors describe the endo-mannanase activity of *RiGH26* and hypothesise on the role of two CBM modules to anchor manno-oligosaccharides to the surface of the cell. The authors also report a new specificity for *RiGH113* as a reducing end mannose-releasing exo-oligomannosidase. The authors also describe a number of enzyme activities involved in β -mannan degradation and go on to predict a model for β -mannan degradation by *R. intestinalis*. The authors also test the ability of *R. intestinalis* to compete with another β -mannan degrader *Bacteroides ovatus* showing that the two can co-exist whilst using β -mannan as a carbon source.

Notably, they observed *R. intestinalis* was able to out compete *B. ovatus* during stationary phase suggesting *R. intestinalis* has a selective advantage during nutrient limitation. Finally the authors showed that in a minimal microbiota grown on β -mannan members capable of breaking down β -mannan are enriched and that mucin degrader levels decreased.

This paper is of significant interest to the field as it details the degradation of complex polysaccharide by the firmicute *R. intestinalis*, most studies of polysaccharide degradation in the human gut have centred on the bacteroides. It provides new insights and adds depth to the body of knowledge which could influence the development of targeted prebiotics. The paper has a significant amount of data including bacterial growths, regulation data, enzyme activities, competition experiments and in vivo experiments. It is clear to see that the paper is a considerable amount of work by the authors.

We thank the referee for positive comments, overall interest in data, and valuable suggestions to improve the manuscript.

Major points to address:

- 1) Fig.2a: The authors show growth curves for *R. intestinalis* on different carbon sources. The figure shown only displays growth curves over 8hrs which only covers growth into exponential phase. Usually, I would expect to see a full growth curve right through to stationary phase. Later in the manuscript the authors show HPLC traces of culture supernatant from stationary phase growths. It would be beneficial to show the full growth curve in Fig2a.

Re: As suggested by the reviewer, an updated Fig. 2a displaying the full growth profile of *R. intestinalis* on different β -mannans is now included. In addition, we have submitted an updated Fig. 2b with pH measurements over the stationary phase with the revised manuscript.

- 2) Supplementary Fig 1a. The chromatogram is shown from 2.5 min. There appears to be peaks that would align with the mannose and oligo standards. The Authors state that there was no observed build-up of oligos or mannose, what are the peaks between 2.5 and 10 min?

Re: Following reviewer's comment, the chromatogram shown in Supplementary Fig. 1a starts now from 0 min. Moreover, we have reduced the

number of time points to improve the figure readability. No peak in the stationary phase culture supernatant aligns with the mannose or manno-oligosaccharide standards. As shown in Supplementary Fig. 1b, the same samples were also run on a system for monosaccharide analysis to improve the separation of the monosaccharide and oligosaccharide peaks. To make clear that there is no accumulation of manno-oligosaccharide, we have moved the traces with the standards on top of the figure, closer to the trace displaying the YCFA+AcGGM from stationary phase. Again, no peak in the stationary phase culture supernatant corresponds with the mannose or manno-oligosaccharide standards. In addition, we have conducted an additional experiment (not intended for publication) and the result is shown in the figure below. In this experiment the YCFA+AcGGM from the stationary phase (blue) has been incubated with the mannosidase *RiGH113* (light blue) for 16 h at 37 °C and the products analyzed by HPAEC-PAD. While *RiGH113* released mannose (light green) from a manno-oligosaccharide control (green), no mannose was released from the YCFA + AcGGM from the stationary phase. These results show that the peak 1-2-3 are not manno-oligosaccharides thus clearly demonstrating that there is neither mannose nor accumulation of manno-oligosaccharides in *R. intestinalis* spent medium after growth on AcGGM.

- 3) Page 10 line 201 onwards, the authors claim that the frequency of galactose substitutions does not affect *RiGH36* catalytic efficiency. From the data shown it is hard to come to that conclusion, the authors show end point assays displaying *RiGH36* is capable of fully hydrolysing oligos with one or two internal substitutions, this does not show how quickly or efficiently the oligos

are hydrolysed. To assess the catalytic activity the authors need to either measure the catalytic parameters of the enzyme for each substrate or a time course showing the oligos are degraded in a similar fashion/speed using the same conditions for both oligos.

Re: We agree with the reviewer that the data on page 10 line 201-206 do not provide evidences that the frequency of galactosylations does not affect *RiGH36* catalytic efficiency. We apologize for the confusion and thank the reviewer for catching this. The goal of the two experiments with galactosylmannotriose and digalactosylmannopentaose was to demonstrate that *RiGH36* is able to catalyze the removal of galactose from the reducing end of a sugar or the removal of two internal consecutive substitutions. As the manuscript contains a great deal of data, we felt it best to focus on the activity and characterization of the enzyme and lay the groundwork for further studies regarding the enzyme kinetics. These aspects will be addressed in another paper.

For clarity purposes, we have reformulated the sentence at lines 200-203 which now says: “Beside cleaving single internal galactose residues from manno-oligosaccharides, this enzyme was capable of removing α -1,6-galactose from the reducing-end of a substituted manno-oligosaccharide (Supplementary Fig. 8e) and from an oligosaccharide containing two consecutive substitutions (Supplementary Fig. 8f)”.

Moreover, info about the concentration of the enzymes and substrates used for the assays have been added to the legend of the Supplementary Fig. 8.

- 4) Finally, the authors show a model of β -mannan degradation based on the assumption *RiGH26* is located on the cell surface and that is the key enzyme which breaks down β -mannan to oligos for transport by the ABC transporter. I think if this is the case the authors need to show the cellular location for *RiGH26* experimentally.

Re: The reviewer brings up a good point and following his/her suggestion we have obtained the polyclonal antibody raised against the recombinant *RiGH26* and conducted immunofluorescence microscopy. The results, shown in Fig. 3 in the revised manuscript, demonstrate that *RiGH26* is located on the surface

of the *R. intestinalis* cells, similarly to the previously shown surface enzyme *RiXyn10A* from the same bacterium (Leth et al., Nat Microbiol, 2018).

We have now added the following text in the results section lines 132-133:

“The extracellular localization of *RiGH26* was corroborated experimentally by immunofluorescence microscopy (Fig. 3)”

In the methods section lines 574-594 and in the figure 3 legend lines 942-946.

Minor points:

- 4) The supplemental figures need to be re-ordered so that they appear in the order they are cited in the text.

Re: We thank the reviewer for this constructive suggestion. We have rearranged the panels depicted in the supplemental figures and reordered the supplemental figure numbers to match the sequence they are cited in the text.

- 5) Also there is no consistency in how the HPLC traces are displayed and labelled. Some have different colors, some are labelled beneath the trace or above. It would be helpful if the authors have a uniform way to display the HPLC traces. Also, some of the traces were hard to see as the plots are too close together and should be spaced out more. For example Supplemental fig 7b it is very difficult to see the gal peak as it overlays with another plot and they both have a similar colour.

Re: Due to the fact that there are more than 37 panels showing nearly 200 HPAEC-PAD traces to fully characterize the enzymes described in this paper, it is not always feasible to keep a uniform way to label the traces. However, following reviewer’s comment, we have now consistently placed the labels beside the HPAEC-PAD traces, unless this creates excessive white space in the figure. Moreover, we have amended all the figures by harmonizing the use of the colors to display standards (mannose and manno-oligosaccharides in green; glucose and cello-oligosaccharides in red; galactose in orange), control experiments (blue) and enzyme-treated substrates (light blue). For clarity purposes, color contrast and distance between the plots have been increased

to improve the visibility of the different chromatograms/ms-spectra.

Reviewers' comments:

Reviewer #1 (Remarks to the Author):

In the present revised manuscript the authors present an extensive study of the enzymatic pathway responsible for beta-mannan metabolism in *Roseburia intestinalis*, while also showing how this group of complex polysaccharides allows this gut commensal to compete in its natural environment. The topic is exciting and novel and the work presented in this manuscript is highly impressive and of excellent quality. The revision has significantly improved from the original submission, though there are still a couple of outstanding and relatively minor issues.

The manuscript however is in places difficult to follow, especially in the Results section, probably due to the high density of the provided experimental data. The authors can easily solve this by explaining "what" the authors wanted to show with their experiments and "how" they decided to do it.

Below, some further issues are highlighted:

I could not find any indication of the source of the strain being the main study object of this study (I imagine is a sequenced strain because is publicly available, but was it sequenced within this study or where it was sourced?).

Line 110; rather than listing GH families, it may be better to say 'predicted glycosyl hydrolases belonging to GH113, GH36 and GH1,

Line 120; 'may be' can be written as 'is' since that part of the sentence already contains the word 'suggesting'.

Line 121/122; the sentence may contain a reference to the next paragraph when the function of the GH26 endomannanase will be revealed.

Line 131; 'regions' should be 'region'

Lines 132-134 and Fig 3; the figure is not particularly insightful due to the absence of a negative control (being the strain grown on glucose and/or a *Roseburia* strain which does not encode this protein). A negative control should therefore be provided or mentioned.

Line 141; 'as much as 50 mg/mL Spruce AcGGM', it may be useful to indicate the amount of time and enzyme it takes to do this.

Line 148; the description 'mannanase from Firmicutes mostly in other members of the' is a bit ambiguous, should it be 'mannanases encoded by Firmicutes belonging to various other members of the'?

Line 154; 'to' should be 'for', and 'with highest affinity on' should be 'with its highest affinity for'

Line 168; 'with degree' should be 'with a degree'

Line 406; nutrients should be nutrient

Line 471; spectro-photochemically should be spectrophotometrically?

Line 498; transcription should be translation

M&M; The author may provide more information on bioinformatic analysis; I could not find details of the procedures and parameters used (other than stating the source of the program or online tools used).

M&M; It was not clear to me why the authors in some cases used His-tag purification and in other cases not (in methods section they state that "where appropriate His-tagged transcription [should be translation!] was prevented by the introduction of one of two stop codons"). Also, if non-His-tagged proteins were purified, how was this performed?

Reviewer #2 (Remarks to the Author):

The authors have included experimental data to show the cellular location of the RiGh26 and additional experiments (to the reviewers) to demonstrate that mannose or manno-ologos do not build up in the culture supernatant. The authors have addressed the comments from the reviewers and have edited the manuscript accordingly.

Subject: Response to reviewers' comments of the manuscript NCOMMS-18-21943-A entitled "The Human Gut Firmicute *Roseburia intestinalis* is a Primary Degradator of Dietary β -Mannans".

We appreciate the valuable comments to our manuscript. We now submit a revised version of NCOMMS-18-21943-A that addresses reviewer #1 concerns. Our point-by-point answers to the comments are dealt with chronologically. Our responses to the comments are in red lettering.

Reviewer #1

In the present revised manuscript the authors present an extensive study of the enzymatic pathway responsible for beta-mannan metabolism in *Roseburia intestinalis*, while also showing how this group of complex polysaccharides allows this gut commensal to compete in its natural environment. The topic is exciting and novel and the work presented in this manuscript is highly impressive and of excellent quality. The revision has significantly improved from the original submission, though there are still a couple of outstanding and relatively minor issues.

Re: We appreciate the reviewer's interest in our data and the very positive comments.

The manuscript however is in places difficult to follow, especially in the Results section, probably due to the high density of the provided experimental data. The authors can easily solve this by explaining "what" the authors wanted to show with their experiments and "how" they decided to do it.

Re: We realize that, as referee 1 points out, due to the high concentration of data, the rationale of analysis may be perceived as a little 'hidden' in some places. In most cases, each part of the results section start with a short sentence indicating the hypothesis we wanted to prove with the experiment and how this was performed. Additionally, the title of the each results subsection summarizes the major results. We have identified some less clear parts and added some sentences to improve clarity in the results section.

To clarify the scope of the comparative genomic analysis, binding experiments with the carbohydrate binding modules and the solute binding protein, we have added the following text at lines 119-121:

“We carried out a comparative genomic analysis to establish the distribution of β -mannans utilization loci equivalent to the identified MULL and MULS in other representative *Roseburia spp.* and Clostridium cluster XIVa members”.

and lines 154-158:

“To investigate the biochemical properties of the two modules, *RiCBM27* and *RiCBM23* were expressed in *E. coli* and their capacities to bind to a range of different soluble cello- and manno-oligosaccharides were evaluated using surface plasmon resonance (SPR).”

and lines 170-173:

“The thermodynamic binding parameters of the ABC-transporter associated solute binding protein, *RiMnBP*, to linear and substituted manno-oligosaccharides were determined using isothermal titration calorimetry (ITC).”

Below, some further issues are highlighted:

1) I could not find any indication of the source of the strain being the main study object of this study (I imagine is a sequenced strain because is publicly available, but was it sequenced within this study or where it was sourced?).

Re: We apologize for this oversight and we thank the reviewer for catching this. This info was provided quite late in the results text, lines 90-91. We have now made clear that our study was conducted using the *R. intestinalis* L1-82 and added the strain name at the introduction line 75. In addition, we added the reference describing the isolation of this bacterium at line 473 and the BioProject accession number at line 497.

2) Line 110; rather than listing GH families, it may be better to say ‘predicted glycosyl hydrolases belonging to GH113, GH36 and GH1.

Re: Done as suggested.

3) Line 120; ‘may be’ can be written as ‘is’ since that part of the sentence already contains the word ‘suggesting’.

Re: Done as suggested.

4) Line 121/122; the sentence may contain a reference to the next paragraph when the function of the GH26 endomannanase will be revealed.

Re: We thank the reviewer for this suggestion. We have added the following text at lines 125-126: “(see later results for *R. intestinalis* β -mannanase *RiGH26*)”

5) Line 131; ‘regions’ should be ‘region’.

Re: Done as suggested.

6) Lines 132-134 and Fig 3; the figure is not particularly insightful due to the absence of a negative control (being the strain grown on glucose and/or a *Roseburia* strain which does not encode this protein). A negative control should therefore be provided or mentioned.

Re: We thank the reviewer for the constructive suggestion. As a negative control, we have now included a picture of *R. intestinalis* L1-82 cells cultured on YCFA supplemented with 0.5% w/v glucose and incubated with polyclonal antibodies raised against *RiGH26*. Cells grown on glucose display a minor fluorescence signal, consistent with the proteomics data indicating that *RiGH26* is expressed at basal levels. We believe now that Figure 3 shows the results that would be expected.

Legend (lines 958-967) has been modified as follows:

“Figure 3. Cellular location of the endomannanase *RiGH26*. a, Fluorescent microscopy images of *R. intestinalis* cells cultured on acetylated galactoglucomannan (AcGGM) or glucose (Glc) and incubated with polyclonal antibodies raised against the recombinant endomannanase *RiGH26*. Glucose-grown cells exhibit a low intensity fluorescence signal; this is consistent with the results of the proteomics data showing that, when the organism is cultured on glucose, *RiGH26* is expressed at basal levels. **b,** Fluorescent microscopy images of *R. intestinalis* cells grown on wheat arabinoxylan (positive control) and incubated with antibodies raised against the known surface endoxylanase *RiXyn10A*. Localization microscopy images are representative data from two biological duplicates.”

7) Line 141; ‘as much as 50 mg/mL Spruce AcGGM’, it may be useful to indicate the amount of time and enzyme it takes to do this.

Re: This info was provided in the supplementary Figure 3, lines 111-113. However, following reviewer’s comment, we have rephrased the sentence at lines 144-147.

“Further analysis indicated that *Ri*GH26 is a potent enzyme as, when used at the concentration of 10 nM, it was able to hydrolyze high concentrations of spruce AcGGM (50 mg/mL) into oligosaccharides in 1 h at standard assay conditions (Supplementary Fig. 3d)”.

8) Line 148; the description ‘mannanase from Firmicutes mostly in other members of the’ is a bit ambiguous, should it be ‘mannanases encoded by Firmicutes belonging to various other members of the’?

Re: Done as suggested.

9) Line 154; ‘to’ should be ‘for’, and ‘with highest affinity on’ should be ‘with its highest affinity for’.

Re: Done as suggested.

10) Line 168; ‘with degree’ should be ‘with a degree’.

Re: Done as suggested.

11) Line 406; nutrients should be nutrient.

Re: Done as suggested.

12) Line 471; spectro-photochemically should be spectrophotometrically?

Re: Done as suggested.

13) Line 498; transcription should be translation.

Re: Done as suggested.

14) M&M; The author may provide more information on bioinformatic analysis; I could not find details of the procedures and parameters used (other than stating the source of the program or online tools used).

Re: The supplementary Figure 2 was generated in April 2017; we have now updated the results to include the bacterial genome sequence data deposited after that date and present in the Integrated Microbial Genomes and the NCBI website on the 16th of December 2018. Supplementary Figure 2 and the corresponding legend have been modified accordingly. The identification of mannan-utilization regulons similar to those of *R. intestinalis* L1-82 was based on searches conducted on the Gene Ortholog Neighbourhood viewer on the Integrated Microbial Genomes website.

These was done using the genes encoding *RiGH26* (ROSINTL182_07683, GenBank ABYJ02000124.1:7167-11129) and *RMEP* (ROSINTL182_05476, GenBank ABYJ02000025.1:3200-4429) as the search homolog. Additionally, searches for MULL and MULS-like proteins were also performed by BLAST against the NCBI non-redundant protein sequence database; sequences with coverage <60% and amino acid similarity <45% were excluded. Details used for the searches have been included in the methods section at lines 714-723:

“Comparative Genomic Analysis. Identification of similar β -mannan catabolic genes in bacteria belonging to the Clostridium XIVA cluster was performed using the Gene Ortholog Neighbourhood viewer on the Integrated Microbial Genomes website (<https://img.jgi.doe.gov>). This was done using the genes encoding *RiGH26* (ROSINTL182_07683, GenBank ABYJ02000124.1:7167-11129) and *RMEP* (ROSINTL182_05476, GenBank ABYJ02000025.1:3200-4429) as the search homolog and the default threshold e-value of 1e-5. Then, a sequence comparison was conducted where each *R. intestinalis* L1-82 RefSeq annotated protein sequence was subjected to BLASTp searches against other Clostridium XIVA members. Sequences with coverage <60% and amino acid similarity <45% were excluded”.

15) M&M; It was not clear to me why the authors in some cases used His-tag purification and in other cases not (in methods section they state that "where appropriate His-tagged transcription [should be translation!] was prevented by the introduction of one of two stop codons"). Also, if non-His-tagged proteins were purified, how was this performed?

Re: We thank the reviewer for this insight. When we started this study, we conducted protein sequence analyses with previously characterized enzymes to identify catalytic and active residues and determine the most suitable cloning strategy. The reasons for having four non-His tagged proteins - *RiGH113*, *RiGH36*, *RMep* and *RiPgm* - are as follows. Based on the sequence similarities with the *Ruminococcus albus* epimerase *RaCE* (PDB ID: 3VW5), we observed that *RMep* has a catalytic histidine (His 395) close to the C-terminus that could have interaction with a potential C-terminal His-tag. Based on the sequence similarity with the *AaManA* sequence from *Alicyclobacillus acidocaldarius* (Zhang Y *et al*, J Biol Chem 2008), *RiGH113* has an active residue (Trp-273) close to the C-terminus that may interact with a potential

C-terminal His-tag. Additionally, the His-tagged version of *RiGH36* and *RiPgm* were inactive.

The four non-His tagged proteins (*RiGH113*, *RiGH36*, *RiMep* and *RiPgm*), were purified using hydrophobic interaction chromatography (HIC) followed by Size Exclusion Chromatography (SEC), as indicated in the Supplementary table 9. The purification methods were already described at methods lines 520-531. We have now made a few minor changes in the methods section for clarification:

a) lines 508-511:

“Recombinant proteins generally contained a C-terminal His₆-tag, although, in some cases, His-tag translation was prevented by the introduction of one or two stop codons at the end of the open-reading frame (*RiMep*, *RiGH36*, *RiPgm* and *RiGH113*).”

b) added the following text at lines 511-512:

“The His₆-tag was excluded to prevent interaction with putative C-terminal active or catalytic residues that could be detrimental to the enzymes' activity”

c) and specified that the proteins *RiMep*, *RiGH36*, *RiPgm* and *RiGH113* were purified by HIC at line 524.